# Multispectral optoacoustic imaging of dynamic redox correlation and pathophysiological progression utilizing upconversion nanoprobes

Xiangzhao Ai [1,2], Zhimin Wang[2], Haolun Cheong[2], Yong Wang [3], Ruochong Zhang[4], Jun Lin[5], Yuanjin Zheng[4], Mingyuan Gao[3] & Bengang Xing[1,2]

Precise and differential profiling of the dynamic correlations and pathophysiological implications of multiplex biological mediators with deep penetration and highly programmed precision remain critical challenges in clinics. Here we present an innovative strategy by tailoring a powerful multispectral optoacoustic tomography (MSOT) technique with a photon-upconverting nanoprobe (UCN) for simultaneous visualization of diversely endogenous redox biomarkers with excellent spatiotemporal resolution in living conditions. Upon incorporating two specific radicals-sensitive NIR cyanine fluorophores onto UCNs surface, such nanoprobes can orthogonally respond to disparate oxidative and nitrosative stimulation, and generate spectrally opposite optoacoustic signal variations, which thus achieves compelling superiorities for reversed ratiometric tracking of multiple radicals under dual independent wavelength channels, and significantly, for precise validating of their complex dynamics and correlations with redox-mediated pathophysiological procession in vivo.

[1] Sino-Singapore International Joint Research Institute (SSIJRI), Guangzhou 510000, China. [2] Division of Chemistry and Biological Chemistry, School of Physical and Mathematical Sciences, Nanyang Technological University, Singapore 637371, Singapore. [3] State Key Laboratory of Radiation Medicine and Protection, School for Radiological and Interdisciplinary Sciences (RAD-X), Soochow University, Suzhou 215123, China. [4] School of Electrical and Electronic Engineering, Nanyang Technological University, 50 Nanyang Avenue, Singapore 639798, Singapore. [5] State Key Laboratory of Rare Earth Resource Utilization, Changchun Institute of Applied Chemistry, Chinese Academy of Sciences, Changchun 130022, China. These authors contributed equally: Xiangzhao Ai, Zhimin Wang. Correspondence and requests for materials should be addressed to M.G. (email: gaomy@iccas.ac.cn) or to B.X. (email: bengang@ntu.edu.sg)

Currently, early disease theranostics in clinics demands the capability to comprehensively understand the intricate signaling pathways in many health-threatening illnesses, and to precisely track their physiological and pathological development in real time[1]. Considering the heterogeneous and complex nature of living systems, the bioassay reporters addressing single biological pathway may not be able to fully reveal the biodiversity. In addition, lack of sufficient sensitivity and specificity to represent multiple pathophysiological variations may greatly restrict their effective validation of disease pathogenesis at the different stages[2]. Development of specific and unified strategies that allow multiplex screening of various biomarkers, and, importantly, to precisely reflect the dynamic correlations of different signaling bioregulators associated with the etiology of diseases and procession remains challenging in the fields.

As an essential signaling mediator in human beings, multiple redox radicals, including reactive oxygen and nitrogen species (ROS/RNS), have been extensively authenticated as significant functional regulators involved in many essential physiological processes such as cellular communication, signal transduction, intermediary metabolism, and immune or inflammatory response[3]. The altered redox balances may cause severe oxidative or nitrosative stress that could be closely implicated in the etiology and pathologies of diverse human diseases[4]. Moreover, mounting investigations have indicated that the generation of ROS or RNS is not static, but rather, their excess or shortage, or even spatiotemporal distributions and correlations are always processing in a highly dynamic and programmed precision. Such biological diversities of free radicals provide great possibilities to act as ideal endogenous biomarkers for spatiotemporally dynamic profiling of the pathophysiological implications in complicated living settings.

Conventional strategies through individual radical sensing encountered technical concerns that may critically prevent their implementation for direct determination of multiple free radicals within a programmed and longitudinal resolution[5–8]. Although monitoring of both oxidative and nitrosative stress could be initially achieved through combination of different sensing moieties[9,10], in vivo imaging of dynamic changes of different redox species orthogonally and real-time tailoring of their close correlations with pathophysiological processing remain challenging. The lack of "smart" and unified tools for concurrent recognition of various radicals in deep-seated tissue is still an obvious impediment, and relevant investigations are thus highly desired.

Recently, the lanthanide-doped upconversion nanocrystals (UCNs) have been extensively applied in biosensing, molecular imaging, and nanomedicine, due to their extraordinary capability to convert near-infrared (NIR) photonic excitations into multiplexed emissions ranging from UV to NIR windows[11–14]. Such unique tissue-penetrable, emission-tunable, and remarkably multiplexing optical properties, featured by a single photonic excitation, can ideally realize a precise interrelation and meet complex biological demands by fitting different sensing moieties into one rationally integrated nanomatrix, thus rendering UCNs a superior multispectral reporter to simultaneously read out numerous analytes (e.g., ROS and RNS) in highly complex and dynamic living environments[15–17].

As an amazing imaging modality, multispectral optoacoustic tomography (MSOT), which can supply reliable anatomy information to the disease theranostics in pre-clinical trials, has recently attracted considerable attention in biomedical sciences[18–20]. MSOT can construct accurate tomographic images in vivo by utilizing non-ionizing NIR radiation to generate broadband ultrasonic waves, which provides promising signal-to-noise ratio and high-resolution exquisite images at depths in living animals that are hardly accessible by conventional optical

imaging approaches[21–23]. Importantly, MSOT demonstrates multi-wavelength option that can be selectively performed to concurrently exploit different absorbing agents with a well-defined spatiotemporal resolution, thus providing more information-rich feasibility to monitor dynamic phenomena through multiple sensing channels, which therefore promote the exploration of biological progressions and theranostic development in clinics[24–27].

In this work, we present an innovative approach for simultaneous screening of various redox species, and, significantly, for dynamic profiling of their intricate correlations with pathophysiological implications by using NIR light-mediated UCNs as optoacoustic (OA) nanoprobes and multispectral signal acquisition through MSOT imaging. By taking advantages of multiplexing luminescence of UCNs converted from NIR laser illumination, two specific ROS- and RNS-sensitive NIR cyanine fluorophores with their absorbance overlapping the upconverted UCNs emissions were individually incorporated onto the nanoprobe surface. Such unique chromogenic modification guaranteed UCNs with a broad absorbance in NIR region and made it a unique probe to monitor the MSOT signal variations. Under oxidative or nitrosative stimulation, the radical oxidations triggered the structural rearrangement or degradation in fluorophores, which thus contributed spectrally opposite MSOT response for reversed ratiometric tracking of multiple radicals with a longitudinal resolution, and, meanwhile, for dynamic profiling of their correlations with the pathogenesis in live mice.

## Results

**Design of UCNs to differentiate disparate redox species.** Here, a rational design by integration of NIR photon-mediated UCNs with a unified MSOT imaging was demonstrated to orthogonally identify disparate oxidative or nitrosative radicals (e.g., $O_2^{\bullet-}$ or $ONOO^-$) in a highly spatiotemporal precision (Fig. 1). Different from conventional probing systems via "on or off" concept for radicals sensing, the rationale of current imaging probes mainly relies on the unique spectral features of two different NIR fluorophores on the UCN surface, which can concomitantly respond to ROS and RNS stimulation to produce OA signals in opposite directions at different wavelengths. Such different redox-triggered spectral changes could be interlocked in a ratiometric precision, providing great benefits for simultaneous MSOT imaging of ROS and RNS dynamics under two independent optical channels.

Typically, a specific NIR light-mediated upconversion platform was fabricated by embedding two ROS/RNS-responsive substrates in a branched polymer layer on the surface of lanthanide-doped UCNs via hydrophobic interaction[28]. Upon ROS treatment, the hydrocyanine substrate (HCy5) regenerated the π-conjugation in Cy5 with maximum absorbance at 640 nm[29], leading to enhanced OA signal at the same wavelength. While upon RNS stimulation, another specific cyanine substrate (Cy7) with maximal absorbance at 780 nm underwent the structural degradation, causing OA signal reduction at 800 nm[30]. Moreover, considering the multicolor emissions of UCNs under 980 nm irradiation, the upconverted luminescence (UCL) decreased at 660 nm with ROS treatment, and increased at 800 nm with RNS incubation via the processes of energy transfer[31]. Such promising imaging strategies through MSOT and UCL provide a multiplex signal readout for different radicals at NIR spectral window, thus facilitating the non-invasive multiple radicals detection within deep tissue in complex living conditions.

**In vitro characterization of UCNs.** The upconversion nano-platforms with multiple emissions at 660 and 800 nm were established by doping core-shell UCNs with $Er^{3+}$ and $Tm^{3+}$

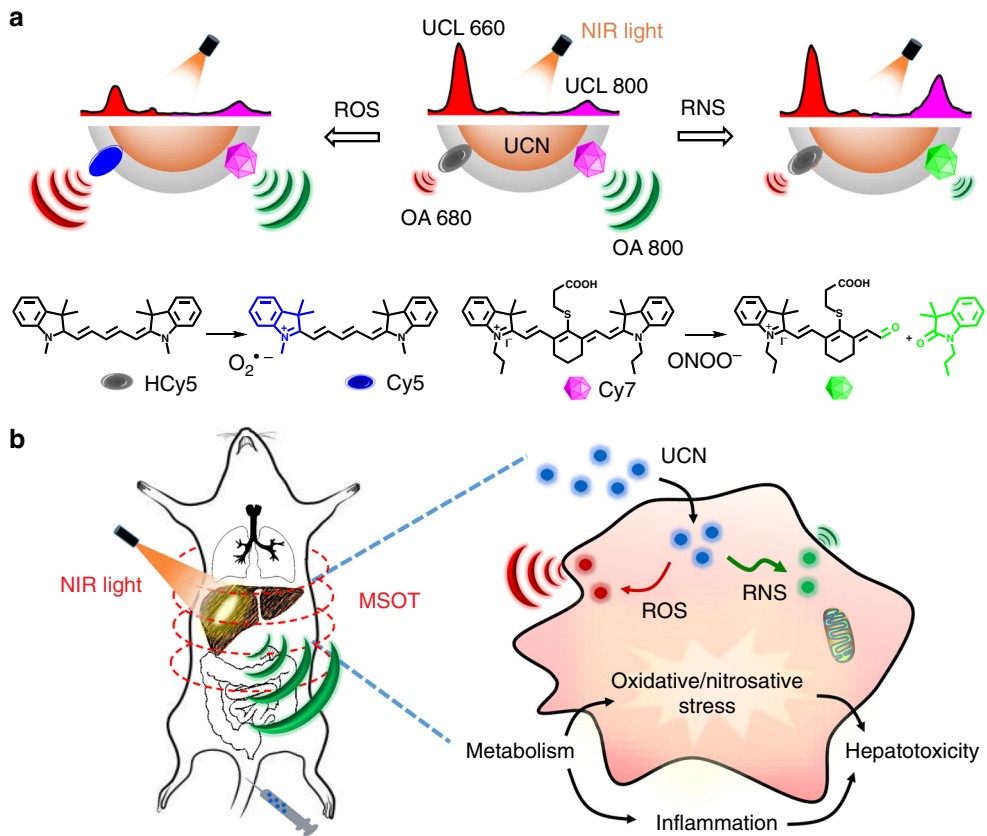

**Fig. 1** Illustration of the multiple radical-sensitive approach for dynamic profiling of pathophysiology implications in vivo. **a** Design of near-infrared (NIR) light-mediated upconversion nanoprobe by incorporating reactive oxygen and nitrogen species (ROS)-responsive HCy5 and reactive nitrogen species (RNS)-responsive Cy7 onto upconversion nanocrystal (UCN) surface. Upon radical stimulation, HCy5 and Cy7 underwent the structural regeneration and degradation, respectively, leading to ratiometric upconverted luminescence (UCL) and optoacoustic (OA) signal variations in NIR spectral region. **b** Dynamic profiling of pathophysiological implications to explore the underlying radical-induced inflammations by tailoring multispectral optoacoustic tomography (MSOT) with optical properties of UCNs in live mice

ions[32]. Transmission electron microscopy (TEM) images indicated a spherical morphology of UCNs with narrow size distribution at ~20 nm (Supplementary Fig. 1). These core-shell nanostructures were further modified with branched polyethylenimine (PEI$_{25000}$) and polyethylene glycol acid (PEG$_{5000}$-COOH) to enhance biocompatibility and to facilitate the incorporation of radical-responsive HCy5/Cy7 into the particle structure (Fig. 2a and Supplementary Fig. 2). Laser irradiation (e.g., at 980 nm) of the polymer-modified UCNs emitted UCL luminescence at 660 and 800 nm (Supplementary Fig. 3), which decreased upon HCy5 or Cy7 coating on the particle surface due to the process of energy transfer between fluorophores and UCNs (Supplementary Figs 3 and 4). The obvious absorbances of cyanine fluorophores were used to determine the optimal loading, and there were 8.9% HCy5 and 12.2% Cy7 found on the UCN surface (Supplementary Fig. 5). The dynamic light scattering (DLS) and zeta potential analysis indicated a uniform hydrodynamic diameter (100 ± 27 nm) and positive charge surface (21.8 ± 2.8 mV) (Fig. 2a and Supplementary Fig. 6), which also exhibited great stability in phosphate-buffered saline (PBS) buffer solution (Supplementary Fig. 7).

We first evaluated the capabilities of UCNs to differentiate multiple radicals in buffers (pH 7.4). Prior to treatment with radicals, UCNs alone presented a strong absorbance of Cy7 at 780 nm (Fig. 2b). In the presence of typical ROS, for example, superoxide anion ($O_2^{\bullet-}$, 100 μM), a significant absorbance enhancement at 640 nm was observed, mainly due to the

HCy5 structural regeneration induced by rapid oxidation (Supplementary Fig. 8), while similar ROS treatment will not alter Cy7 structure in UCNs, and there was no obvious spectral change at 780 nm, suggesting the selective $O_2^{\bullet-}$ recognition between HCy5 and Cy7. Conversely, in the presence of RNS, for example, peroxynitrite ($ONOO^-$, 100 μM), almost no absorbance change at 640 nm, but progressive absorbance decrease at 780 nm was detected on UCNs due to the irreversible oxidative degradation of Cy7 structure (Fig. 2b and Supplementary Fig. 9). Moreover, the combined stimulation of UCNs with ROS (e.g., $O_2^{\bullet-}$, 20 μM) and RNS (e.g., $ONOO^-$, 20 μM) led to the selective absorbance increase at 640 nm and decrease at 780 nm simultaneously (Fig. 2b), suggesting that the opposite absorbance change triggered by specific radicals would provide unique advantages for ratiometric mapping of the ROS/RNS dynamics in living settings. Furthermore, apart from a certain similarity between $O_2^{\bullet-}$ and $^{\bullet}OH$, as well as the oxidative Cy7 degradation between $ONOO^-$ and $OCl^-$, no obvious response was observed upon other radical treatment, including $ROO^{\bullet}$, NO, and $H_2O_2$, and so on (Fig. 2c), clearly showing the beneficial specificity of UCNs for simultaneous sensing of ROS (e.g., $O_2^{\bullet-}$) and RNS (e.g., $ONOO^-$) separately.

We also examined the capability of UCNs as OA nanoprobes to differentiate ROS/RNS species under physiological conditions. Typically, we incubated UCNs with $O_2^{\bullet-}$ and $ONOO^-$ in PBS (pH 7.4). Then, the radical-triggered OA changes were collected by a unique MSOT setup with NIR excitation at the range of

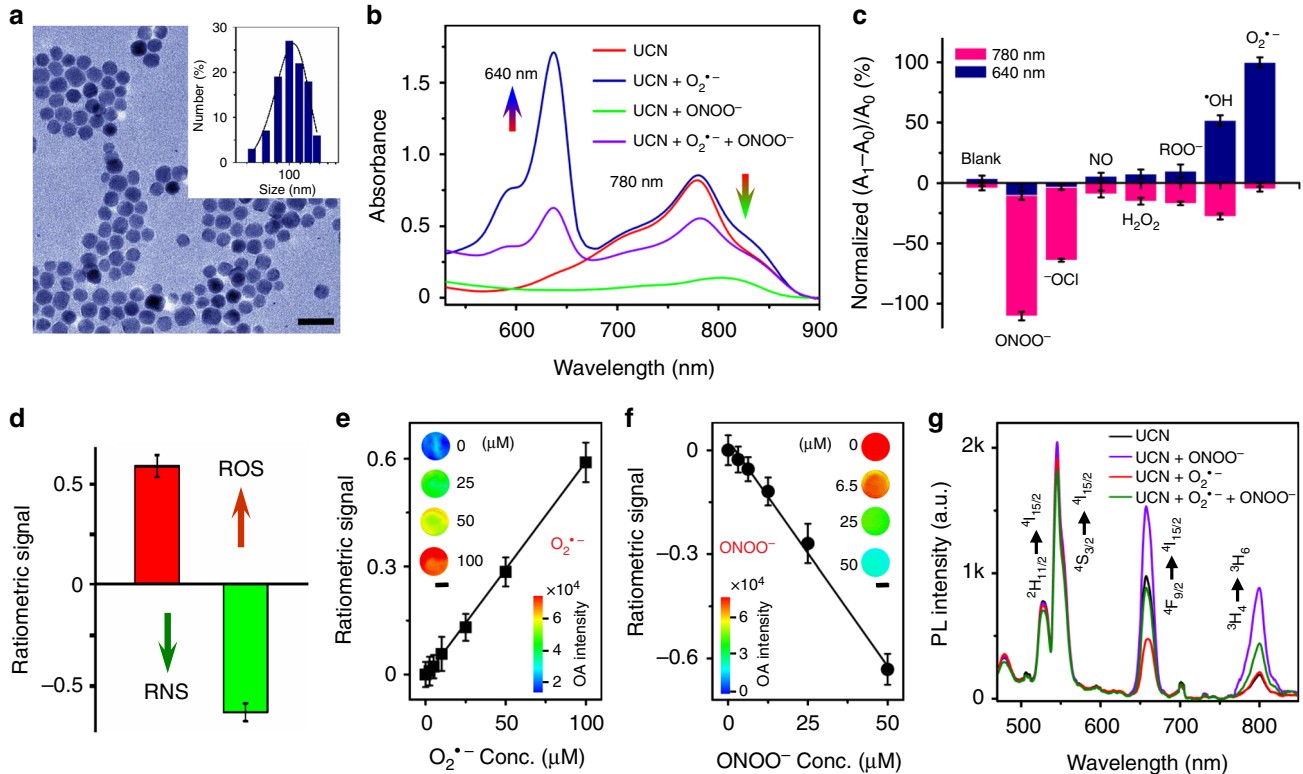

**Fig. 2** Characterization of optoacoustic (OA) and upconverted luminescence (UCL) signals of upconversion nanocrystals (UCNs) upon reactive oxygen and nitrogen species (ROS/RNS) treatment in buffers. **a** Transmission electron microscopy (TEM) and dynamic light scattering (DLS) analysis of UCNs. Scale bar: 50 nm. **b** Ultraviolet–visible (UV–vis) spectra of UCNs (1 mg mL$^{-1}$) upon ROS/RNS treatment. **c** Specificity of UCNs with the normalized absorbance at 640 and 780 nm upon various radicals (100 μM) stimulation. **d** Deconvoluted ratiometric OA signals in the absence (OA$_0$) or presence (OA$_1$) of ROS (O$_2^{\bullet-}$) and RNS (ONOO$^-$). Ratiometric analysis at 680 and 800 nm: ($\Delta$OA$_{680}$ + $\Delta$OA$_{800}$)/OA$_{800}$. **e, f** Ratiometric OA signal changes upon different concentration of O$_2^{\bullet-}$ (**e**) or ONOO$^-$ (**f**) treatment. Scale bar: 1 mm. **g** UCL spectra of UCNs upon O$_2^{\bullet-}$ or ONOO$^-$ stimulation (Ex: 980 nm). Data were represented as mean ± standard deviation (SD)

680–980 nm, which were further processed through mathematical deconvolution for specific differentiation of ROS and RNS response at different wavelengths[33]. As shown in Fig. 2d and Supplementary Fig. 10, UCNs treated with O$_2^{\bullet-}$ demonstrated significant OA signal enhancement at 680 nm, but negligible change at 800 nm, leading to a ratiometric value (($\Delta$OA$_{680}$ + $\Delta$OA$_{800}$)/OA$_{800}$) of 0.59 ± 0.06. However, the addition of ONOO$^-$ obviously decreased the maximum OA signal at 800 nm rather than that at 680 nm, thus contributing a dramatic reverse ratiometric change (($\Delta$OA$_{680}$ + $\Delta$OA$_{800}$)/OA$_{800}$ = −0.63 ± 0.04) (Fig. 2d). Such opposite OA signal changes supplied an ideal amenability to dual-channel sensing of ROS and RNS at 680 and 800 nm, respectively[34]. The OA response was also linearly correspondent to different radicals in buffers with sensing limit down to 85 and 168 nM for O$_2^{\bullet-}$ and ONOO$^-$, respectively (Fig. 2e, f). Moreover, considering the complexities of multiple radical generation and distribution in vivo, the oxidative response of simultaneous radicals was also evaluated by the multiplexing UCL signals upon NIR irradiation of UCNs at 980 nm. As shown in Fig. 2g, stimulation of UCNs with O$_2^{\bullet-}$ led to a reduced UCL emission at 660 nm due to the energy transfer between UCNs and regenerated Cy5 triggered by ROS oxidation. However, ONOO$^-$ treatment degraded Cy7 structure on the UCN surface, which therefore caused the recovery of UCL emissions at 660 and 800 nm, respectively. To mimic the clinical skin penetration, we examined the UCL and OA signals with pork tissues at different thickness. An 8-mm-thick tissue significantly reduced the UCL emission at 660 nm (~75%), while only ~16% drop at 800 nm after 980 nm irradiation. Notably, only the minor OA

attenuations were presented at 680 nm (~7% decrease) and 800 nm (~4% decrease) in MSOT imaging with negligible ratiometric variations, suggesting a superior penetration depth achieved by MSOT as compared to UCL imaging (Supplementary Fig. 11). The consistent MSOT and UCL analysis demonstrated the great feasibility of UCNs as promising multimodality nanoprobes for simultaneous imaging of oxidative and nitrosative stresses in complex living conditions.

**UCN response to endogenous redox species in live cells**. We further investigated the capability of UCNs to monitor the endogenously generated radicals in murine RAW264.7 macrophage cells through both MSOT and optical UCL imaging. Briefly, the excessive O$_2^{\bullet-}$ and ONOO$^-$ generation in cells was achieved by pretreating with phorbol 12-myristate 13-acetate (PMA) and a mixture of lipopolysaccharide (LPS)/interferon-γ (INF-γ)/PMA, respectively (Fig. 3a). Upon confirmation of cellular radical stimulation by standard fluorimetric peroxynitrite (green) and superoxide (red) assay (Supplementary Fig. 12)[6,35], the cells were collected for MSOT measurement after 4 h incubation with UCNs. As shown in Fig. 3b and Supplementary Fig. 13, cellular inflammation triggered by PMA produced O$_2^{\bullet-}$ species (group 3), which presented an obvious OA enhancement at 680 nm, but less change at 800 nm, leading to a dramatic ratiometric signal (($\Delta$OA$_{680}$ + $\Delta$OA$_{800}$)/OA$_{800}$) change with a value of 0.25 ± 0.03. However, the cell stimulation with LPS/IFN-γ/PMA for ONOO$^-$ generation (group 5 in Fig. 3c) displayed an obvious attenuation at 800 nm, but less change at 680 nm,

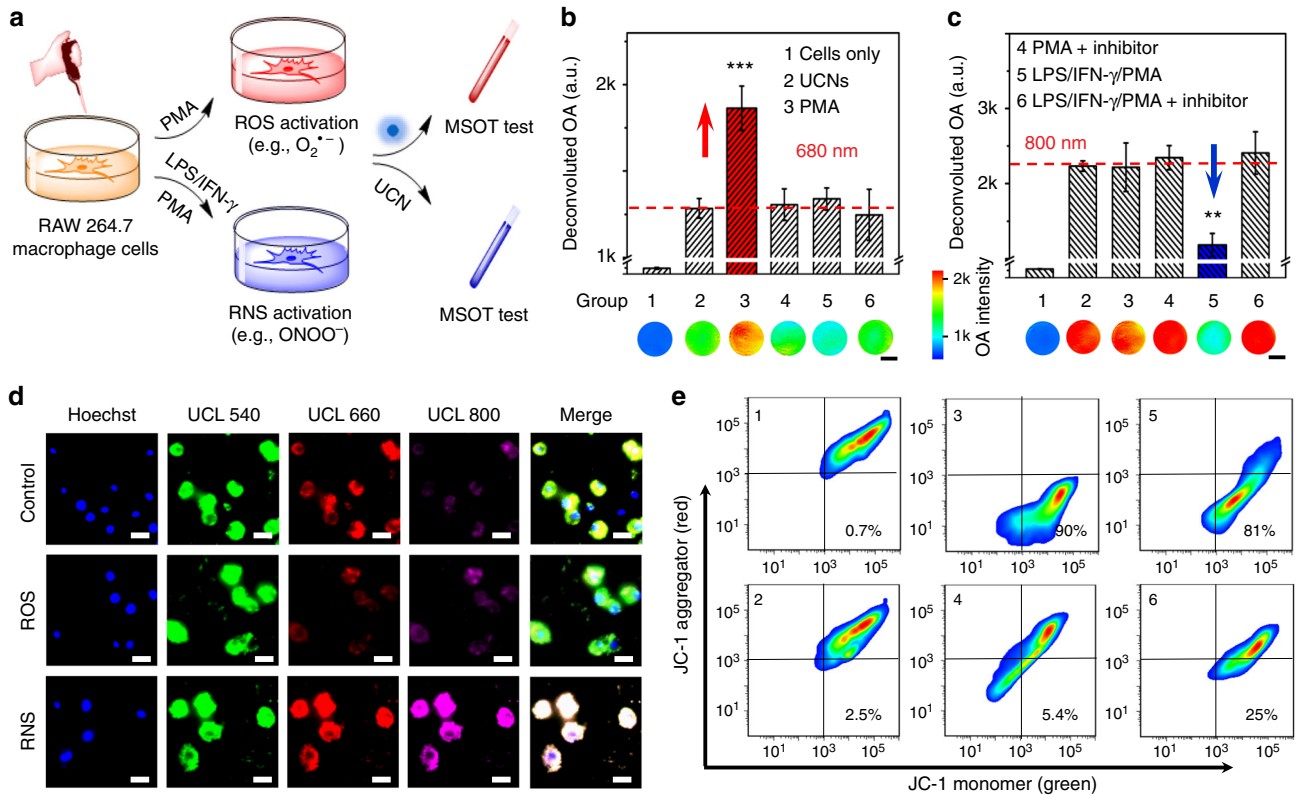

**Fig. 3** Inflammatory response of endogenous redox species by upconversion nanocrystals (UCNs) in live cells. **a** Scheme of endogenous reactive oxygen and nitrogen species (ROS/RNS) stimulation and inflammation profiling through UCNs in RAW264.7 macrophage cells. **b, c** Deconvoluted optoacoustic (OA) signals at 680 nm (**b**) and 800 nm (**c**) of macrophage cells in different groups: (1) cells only; (2) cells incubate with UCNs; (3) phorbol 12-myristate 13-acetate (PMA)-stimulated cells ($O_2^{\bullet-}$ generation) with UCNs; (4) PMA-stimulated cells with $O_2^{\bullet-}$ inhibitor (Mn(III) tetrakis (4-benzoic acid) porphyrin (MnTBAP)) and UCNs; (5) lipopolysaccharide (LPS/interferon-γ (INF-γ)/PMA-stimulated cells (ONOO$^-$ generation) with UCNs; (6) LPS/INF-γ/PMA-stimulated cells with ONOO$^-$ inhibitor (mercaptoethyl guanidine (MEG)) and UCNs. Scale bar: 1 mm. **p** < 0.01, ***p < 0.001. Data were represented as mean ± SD. **d** Upconverted luminescence (UCL) fluorescence imaging of RAW264.7 cells incubated with UCNs (100 μg mL$^{-1}$) in the absence (top), in the presence of $O_2^{\bullet-}$ (middle), and ONOO$^-$ (bottom) generation. Blue: Hoechst 33342 (Ex: 405 nm, Em: 460/50 nm), green: UCL-540 (Ex: 980 nm, Em: 540/50 nm), red: UCL-660 (Ex: 980 nm, Em: 640/50 nm), violet: UCL-800 (Ex: 980 nm, Em: 790/30 nm). Scale bar: 20 μm. **e** Flow cytometry (FCM) analysis of mitochondrial membrane potential ($\Delta\psi_m$) by JC-1 in macrophage cells at different groups. Green channel (monomer): Ex: 488 nm, Em: 530/50 nm. Red channel (aggregator): Ex: 561 nm, Em: 610/75 nm

resulting in a reverse ratiometric signal of $-0.83 \pm 0.03$. Additionally, cellular treatment with a scavenger (e.g., Mn(III) tetrakis (4-benzoic acid) porphyrin (MnTBAP) for $O_2^{\bullet-}$ or mercaptoethyl guanidine (MEG) for ONOO$^-$) caused a less OA change with the ratiometric value (e.g., $0.06 \pm 0.02$ and $0.05 \pm 0.01$) similar to the cells under resting states. The UCL signal changes corresponding to ROS/RNS treatment were also examined by confocal microscopy upon 980 nm excitation (Fig. 3d). A progressive loss of UCL emissions was observed at 660 nm with PMA stimulation, indicating the energy transfer between UCNs and regenerated Cy5 after ROS oxidation, while the cellular treatment with LPS/IFN-γ/PMA led to the ONOO$^-$ oxidation of RNS-responsive Cy7 degradation and thus enhanced the UCL emission at 800 nm. Both the radical-responsive UCL changes could be inhibited by $O_2^{\bullet-}$ and ONOO$^-$ scavengers (Supplementary Figs 14 and 15). Moreover, the cell viability studies showed negligible cytotoxicity after incubation with UCNs (0.1 mg mL$^{-1}$) for 24 h (Supplementary Fig. 16). The significant ratiometric OA and UCL signals could easily characterize resting and stress cells, demonstrating the feasibility of UCNs to selectively differentiate the ROS and RNS generation in different cell environments.

We also explored the potential mechanism of ROS/RNS-induced pathology in live cells by determining the mitochondrial membrane potential ($\Delta\psi_m$) with a commercial kit (JC-1), which

mainly keeps as aggregates with red fluorescence in healthy cells, but monomers with green fluorescence after mitochondrial damage[36,37]. The flow cytometry results showed that the macrophage cell population is mainly located at the lower-right quadrant after ROS (group 3) and RNS (group 5) production (Fig. 3e). As a control, negligible signals were determined in the presence of UCNs (group 2), scavenger of $O_2^{\bullet-}$ (group 4), and ONOO$^-$ (group 6), respectively, suggesting the association of over-produced radicals with the mitochondrial dysfunction and initial cell damage in pathological processes.

**Simultaneous screening of multiple radical dynamics in vivo.** The in vivo MSOT and UCL imaging were performed for simultaneous screening of ROS and RNS dynamics via intravenous (i.v.) injection of UCNs (5 mg mL$^{-1}$ in 100 μL saline) into live Balb/c nude mice ($n = 5$). The OA and UCL imaging were captured to monitor the distribution of UCNs in live mice, which consistently showed the obvious particle accumulation in the liver (Supplementary Figs 17–19), suggesting the feasibility of UCN nanoprobes to real-time correlate the radical dynamics in hepatocytes with various stages of inflammatory progression. As a proof of concept, LPS, a bacterial endotoxin from Gram-negative pathogens, which can induce ROS over-production in vivo (e.g.,

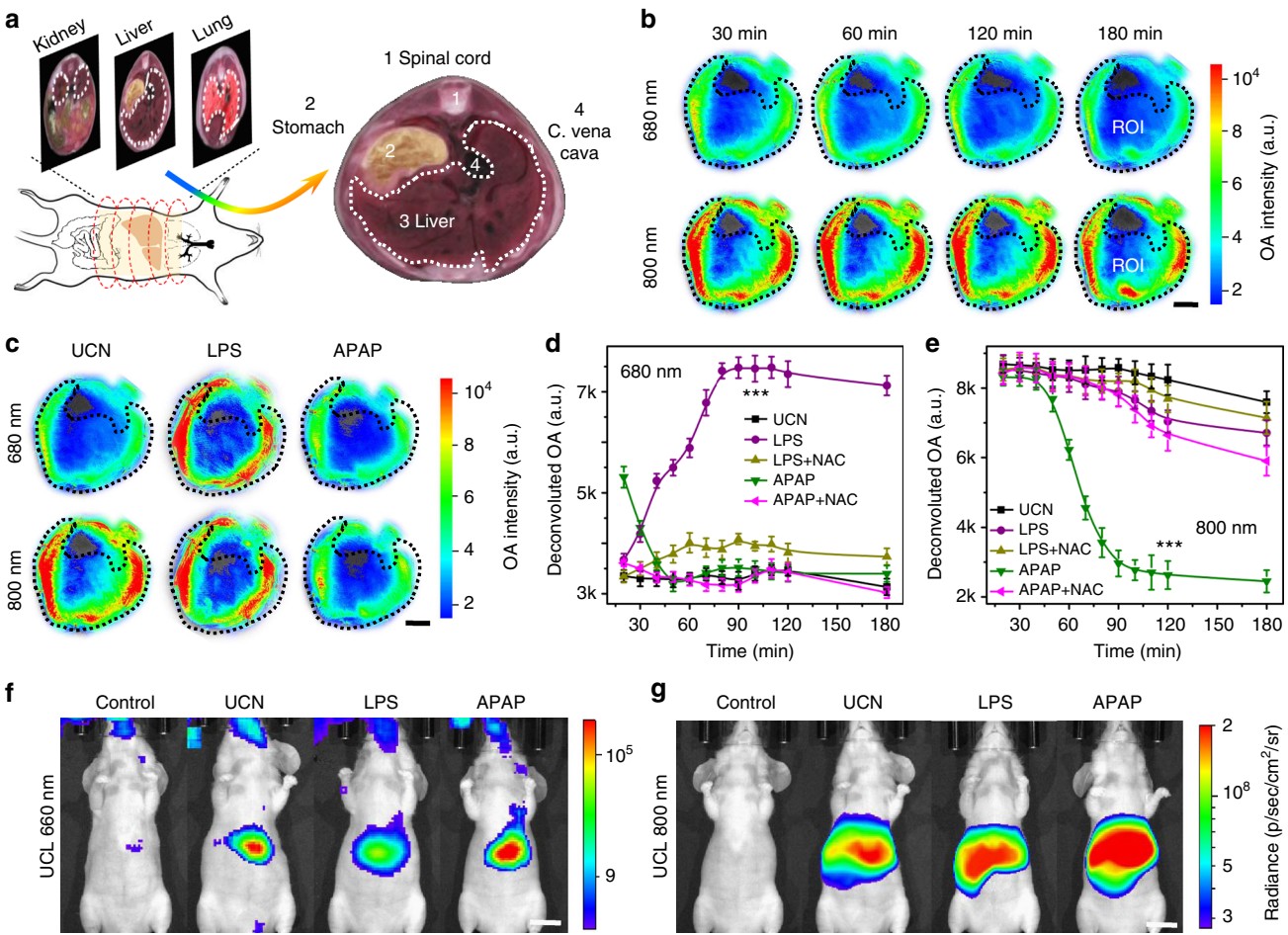

**Fig. 4** Simultaneous profiling of multiple radicals in inflammation models. **a** Scheme of multispectral optoacoustic tomography (MSOT) imaging in abdominal regions of live mice (left) and anatomical image of a liver cross-section from the iThea software (right). **b** Time-resolved MSOT signals at 680 and 800 nm in region of interest (ROI) of liver tomographic images with pseudo-color upon upconversion nanocrystal (UCN) injection ($n = 5$). **c** Optoacoustic (OA) images at 680 and 800 nm in ROI of liver upon UCN, lipopolysaccharide (LPS), and acetaminophenol (APAP) administration for 90 min ($n = 5$). Scale bar: 5 mm. **d**, **e** Dynamic profiling of deconvoluted OA signal variations in hepatic inflammation models by UCNs at 680 nm (**d**) and 800 nm (**e**) upon LPS, APAP, and their reactive metabolite scavenger (NAC) treatment ($n = 5$). Statistical significance was assessed by a Student's $t$ test (heteroscedastic, two-sided). ***$p < 0.001$. Data were represented as mean ± SD. **f**, **g** Representative in vivo upconverted luminescence (UCL) imaging at 660 nm (**f**) and 800 nm (**g**) upon saline, UCN, LPS, and APAP treatment for 90 min (Ex: 980 nm, $n = 5$). Scale bar: 1 cm

$O_2^{\bullet-}$) or acetaminophenol (APAP), a typical anti-pain/fever drug, to stimulate excessive RNS (e.g., $ONOO^-$), were intraperitoneally (i.p.) injected into the mice to mimic early-stage inflammation[38]. Upon subsequent tail-vein administration of UCNs, the MSOT imaging was performed at different time intervals, and the anatomy of OA signals at the region of interest (ROI) in liver cross-sections was evaluated at 680 and 800 nm with pseudo-color processing (Fig. 4a) and mathematical deconvolution (Fig. 4d, e)[33]. As shown in Fig. 4b, the radical stimulation showed negligible OA changes at both 680 and 800 nm over the course of imaging at initial 120 min post injection of UCNs alone, suggesting the reliable baseline and great stability of UCNs in the liver. Nevertheless, LPS stimulation presented a significant enhancement at 680 nm and minimal change at 800 nm with a ratiometric value of 0.42 ± 0.08 at 90 min, while APAP treatment led to an obvious signal attenuation at 800 nm, but negligible change at 680 nm with a reverse ratiometric value of −1.88 ± 0.09 at 90 min after UCN injection (Fig. 4c).

Such redox-responsive ratiometric OA changes and UCL imaging were also used to monitor the dynamic processing of ROS/RNS in LPS/APAP-triggered inflammation animals after tail-vein injection of UCNs. Typically, LPS stimulation resulted

in a gradual ROS increase in mouse liver over the imaging process (Fig. 4d). As compared to the signal variations in the control group, APAP treatment displayed slight OA enhancement at 680 nm related to ROS production (e.g., $O_2^{\bullet-}$) at initial injection of UCNs. However, a dramatically decreasing trend towards ROS response occurred, and continuous $ONOO^-$ increment was easily observed within 3 h of APAP administration (Fig. 4e), suggesting the dynamic correlation between ROS and RNS, and the process of RNS generation was later than that of ROS. When N-acetyl-cysteine (NAC), a glutathione precursor well known to scavenge reactive metabolites in hepatocyte cells[39], was used to treat live mice together with LPS or APAP stimulation, both the OA signals at 680 and 800 nm returned to the levels close to the control group done by UCNs itself, suggesting the effective inhibition of radical over-production by NAC. Meanwhile, the UCL signal also showed an obvious drop (~4.4-folds) at 660 nm in LPS-treated mice and a significant increment (~4.7-folds) at 800 nm in APAP-stimulated mice owing to energy transfer process (Fig. 4f, g and Supplementary Fig. 20), further confirming the great potential of UCNs as a promising nanoprobe for ratiometric screening of radical dynamic in vivo.

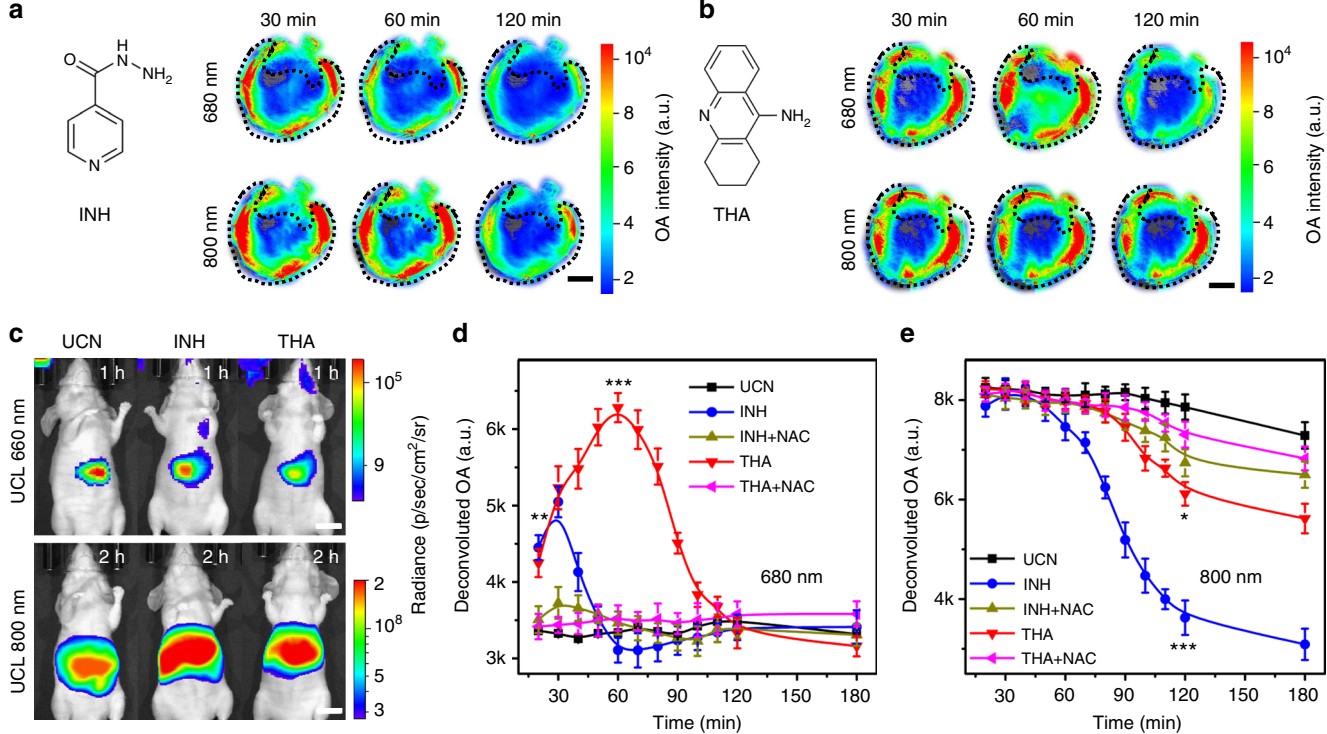

**Fig. 5** Dynamic profiling of multiple radicals variations in the liver upon hepatotoxic drugs treatment. **a**, **b** Time-resolved multispectral optoacoustic tomography (MSOT) signals from upconversion nanocrystal (UCN) at 680 and 800 nm in region of interest (ROI) of liver tomographic images upon isoniazid (INH) (200 mg kg$^{-1}$) and tacrine (THA) (30 mg kg$^{-1}$) treatment ($n = 5$). Scale bar: 5 mm. **c** In vivo upconverted luminescence (UCL) imaging of UCN at 660 and 800 nm upon INH and THA treatment (Ex: 980 nm, $n = 5$). Scale bar: 1 cm. **d**, **e** Dynamic profiling of deconvoluted optoacoustic (OA) signals based on UCN in mouse liver at 680 nm (**d**) and 800 nm (**e**) upon INH, THA, and their reactive metabolite scavenger (NAC) treatment ($n = 5$). Statistical significance was assessed by a Student's $t$ test (heteroscedastic, two-sided). *$p < 0.05$, **$p < 0.01$, and ***$p < 0.001$. Data were represented as mean ± SD

**In vivo monitoring of radical dynamics in pathological settings.** Inspired by in vivo results in standard inflammation animal models for redox species screening, it was highly essential to investigate the heterogeneity of multiple radicals in inflammation development, and, importantly, to correlate their dynamic changes with the pathological progression in living subjects. To this end, isoniazid (INH), a most commonly used anti-tuberculosis drugs[40], and tacrine (THA), a Food and Drug Administration (FDA)-approved agent for Alzheimer's disease treatment[41], have been chosen to mimic the different inflammation stages. INH and THA can induce oxidative or nitrosative stress, which are associated with severe hepatotoxicity in clinics, while the detailed mechanisms have not been fully elucidated[42]. By using UCNs as nanoprobes, we performed a dynamic MSOT imaging of ROS/RNS in the liver upon overdose of INH and THA administration. As shown in Fig. 5a, INH treatment (e.g., 200 mg kg$^{-1}$) presented a short-lived oxidative burst at 680 nm within 30 min (($\Delta OA_{680} + \Delta OA_{800})/OA_{800} = 0.19 \pm 0.08$) and an obvious nitrosative stress-induced attenuation at 800 nm within 120 min (($\Delta OA_{680} + \Delta OA_{800})/OA_{800} = -1.26 \pm 0.07$) after UCN injection. Similarly, the UCL imaging in INH-treated mice also displayed a noticeable emission reduction at 660 nm and an enhancement (~2.9-folds) at 800 nm upon 980 nm excitation (Fig. 5c and Supplementary Fig. 21). The consistent imaging results (Fig. 5d, e) suggested a threshold dose-type generation of ROS (e.g., O$_2^{\bullet-}$) at an early stage, but a dose-dependent excessive RNS (e.g., ONOO$^-$) generation at prolonged duration. Unlike INH, THA administration (30 mg kg$^{-1}$) exhibited a sustained OA increment at 680 nm within 60 min (($\Delta OA_{680} + \Delta OA_{800})/OA_{800} = 0.33 \pm 0.06$), while a slight reverse variation at 800 nm up to 120 min (($\Delta OA_{680} + \Delta OA_{800})/OA_{800} = -0.35 \pm 0.09$)

(Fig. 5b). Furthermore, the UCL imaging displayed considerable emission drop (~3.7-folds) at 660 nm, but less change at 800 nm (Fig. 5c), demonstrating more predominant ROS production than that of RNS in the liver upon overdose of THA treatment.

**Exploring metabolic mechanism and inflammation procession.** It has been established that redox species in living systems closely link to a variety of pathophysiological events from intrinsic metabolism to acute inflammation, and they can also directly reflect the etiological processing of many chronic diseases (Fig. 6a)[3]. In line with the dynamic ROS and RNS changes in vivo, the unique redox-responsive ratiometric MSOT imaging at 680 and 800 nm could serve as an ideal strategy to real-time track the changes of multiple radicals, and to investigate the correlations between ROS/RNS variations and inflammation development in highly complicated living conditions (Fig. 6b). We first examined the inflammation procession in the liver by monitoring two typical metabolism-mediated enzymes activities: cytochrome P450 (CYP450), one major redox source for most phase I drug metabolism, and uridine 5′-diphospho-glucuronosyl transferases (UGTs), a key player for phase II glucuronidation to facilitate the drug biotransformation in vivo[43,44]. As shown in Fig. 6c, d, gradual production of CYP450 and rapid consumption of UGTs were observed at the incipient 60 min after individual administration of APAP, INH, and TNH, indicating that the intensive drug metabolism occurred in the liver. As a model hepatotoxin, APAP mainly undergoes phase II metabolism pathway before its excretion via glucuronidation and sulfation. Only a small proportion of CYP450-dependent phase I metabolism was presented to produce a metabolic iminoquinone, *N*-

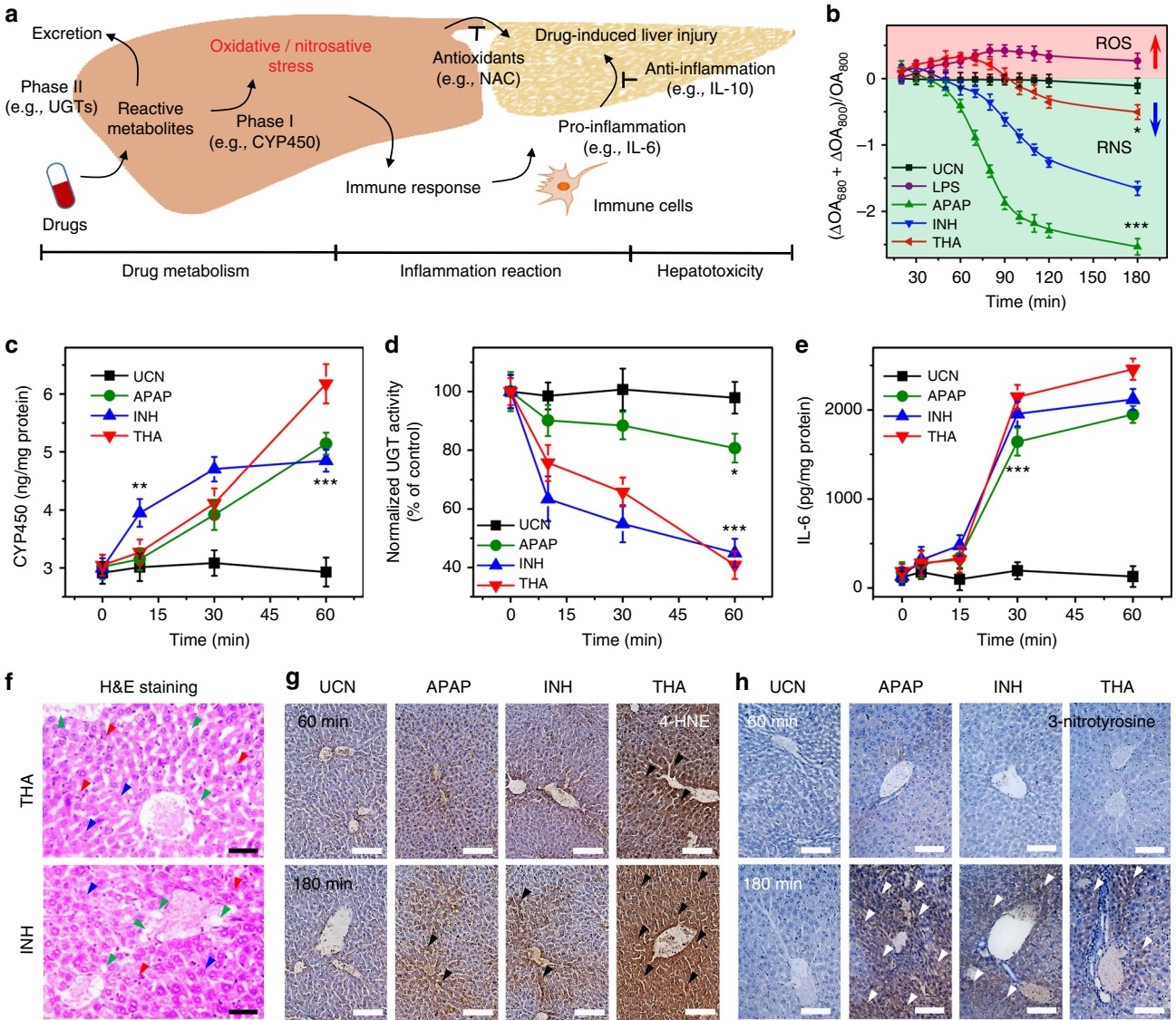

**Fig. 6** Exploration the underlying pathophysiological implications in redox-mediated inflammations in vivo. **a** Illustration of pathological progression of drug-induced liver injury during metabolism and inflammation reaction. **b** Deconvoluted ratiometric optoacoustic (OA) values upon different drug treatment in vivo. **c**, **d** Assessment of metabolism-related enzymes including cytochrome P450 (CYP450) (**c**) and 5′-diphospho-glucuronosyl transferases (UGTs) (**d**) at different times in mouse liver upon drug treatment. **e** Time-resolved variations of inflammation-associated cytokines (interleukin-6 (IL-6)) upon hepatotoxic drug administration in vivo. Statistical significance assessed by a Student's $t$ test (heteroscedastic, two-sided). $*p < 0.05$, $**p < 0.01$, and $***p < 0.001$. **f** Hematoxylin and eosin (H&E) staining in liver tissues at 180 min after isoniazid (INH) and tacrine (THA) treatment ($n = 5$). Arrowheads mark centrilobular vein fibrosis (blue), swollen hepatocytes (green), and inflammatory infiltration (red), respectively. CV: central vein. Scale bars: 50 μm. **g**, **h** Immunohistochemical analysis of 4-hydroxynonenal (4-HNE) (**g**) and 3-nitrotyrosine (**h**) staining in liver section at 60 and 180 min after acetaminophenol (APAP), INH, and THA treatment ($n = 5$). Arrowheads mark 4-HNE-positive lesions (black) and nitrotyrosine-positive foci (white), respectively. Scale bars: 100 μm. Data were represented as mean ± SD

acetyl-$p$-benzoquinone imine, which induced the mitochondrial dysfunction and subsequently initiated nitrosative stress in overdose[42].

Compared to APAP, a similar CYP450 level along with a dramatic depletion of UGT activities was determined upon INH treatment for 60 min (Fig. 6d), suggesting that the metabolism of INH in vivo would mainly go through phase II pathway, while a fraction of INH oxidation via phase I metabolism led to initial radical generation including ROS/RNS, as indicated in ratiometric MSOT analysis (Fig. 6b). Additionally, different from APAP, more significant CYP450 enhancements and UGT suppression were observed upon THA stimulation for 60 min (Fig. 6c, d), indicating the possibility of both phase I and II

pathways occurred in THA metabolism, and the higher phase I metabolism could induce more oxidation of THA for radical generation (e.g., ROS) (Fig. 6b). These data corroborated the controversial metabolic mechanism that THA would be mainly excreted through the glucuronidation process in phase II pathway, and the relevant THA bioactivation involved the CYP450-mediated oxidation to form 1-hydroxytacrine, a reactive metabolite that caused obvious hepatotoxicity with an increasing radical production[45].

Typically, upon drug stimulation, the innate immune responses will initialize the liver tissue injury and repair by releasing the hepatotoxic pro-inflammatory cytokines (e.g., interleukin-6, IL-6) and the hepatoprotective anti-inflammatory cytokines (e.g.,

interleukin-10, IL-10)[46], which will provide valuable insights on real-time investigation of different inflammation stages in vivo. As shown in Fig. 6e and Supplementary Fig. 22, the lower expression levels of IL-6 and IL-10 were observed right after different hepatotoxin injection. However, a significant increase of these cytokines was detected ~60 min post injection, and such high expression remained up to 120 min with the continuous inflammation upon drug stimulations, indicating the presence of both pro- and anti-inflammation that correlated with the radical production upon drug stimulation. Notably, the animal treatment with APAP, INH, or THA for 60 min indicated comparable cytokine levels in the body, while the ratiometric MSOT imaging of the dynamic redox changes within 60 min drug treatment showed completely different profiling (Fig. 6b), showing that the ratiometric MSOT ROS/RNS imaging could precisely differentiate the early inflammation in vivo.

Moreover, we also performed the liver tissue histological and immunohistochemical analysis at 60 and 180 min to evaluate the inflammatory response after animals were treated with different drugs. H&E) staining indicated a minimum hepatotoxicity within 60 min upon treatment with APAP, INH, and THA (Supplementary Fig. 23); however, the typical lobular hepatocyte structures were dramatically destroyed and the disparate histological changes, such as centrilobular vein fibrosis, swollen hepatocytes, sinusoidal congestion, and inflammatory infiltration, were readily observed up to 180 min after drug injection (Fig. 6f and Supplementary Fig. 24). Such severe hepatotoxicity was further confirmed by alanine transaminase (ALT) and aspartate aminotransferase (AST) analysis, both of which are standard biomarkers in clinics for the diagnosis of liver diseases (Supplementary Figs 25 and 26). These studies suggested that the definite hepatotoxicity occurred later than the systematic drug metabolism and initial inflammation reaction, which is the process well accepted in drug-induced livery injury (DILI) (Fig. 6a)[42].

In order to validate the correlations between the dynamic redox changes and inflammation procession in DILI, a specific hallmark of ROS over-production in vivo, 4-hydroxynonenal (4-HNE) assay, was used to examine the oxidative-mediated cell damage. Moreover, as a product of tyrosine nitration in protein, 3-nitrotyrosine was considered as another specific biomarker of nitrosative stress in vivo. As shown in Fig. 6g, h, negligible histological changes in 4-HNE and 3-nitrotyrosine were observed in liver tissues before and after APAP or INH treatment within 60 min. However, slight production of 4-HNE-positive foci and obvious generation of nitrotyrosine-positive lesion were detected at 180 min, indicating the occurrence of both oxidative and nitrosative stress after administration of APAP or INH. Furthermore, both APAP and INH stimulation caused more significant 3-nitrotyrosine staining in the liver ($n = 5$), demonstrating the major roles of RNS (e.g., $ONOO^-$) in the observed inflammation.

Unlike APAP or INH, as an acute hepatotoxic drug withdrawn by FDA[41], THA stimulation led to fewer nitrosative-positive foci, while more 4-HNE-positive lesions were observed at 60 and 180 min, indicating that the oxidative stress may predominantly contribute to the THA-induced liver injury. This histological analysis was in good accordance with the ratiometric MSOT analysis acquired at initial 180 min (Fig. 6b), suggesting the close correlations of ROS/RNS during the DILI processes of each drug, that is, the APAP or INH treatment in mice may induce the quick burst of ROS and high conversion to RNS, while THA-treated mice will undergo more ROS but minor RNS generation. These results demonstrated the great potential of UCN as a multispectral nanoprobe to real-time monitor the dynamic redox changes and correlations of various ROS/RNA biomarkers, and,

importantly, to precisely map the different inflammation stages at initial hepatotoxic conditions.

## Discussion

As essential messenger regulators, ROS and RNS are tightly associated with various pathophysiological processes, ranging from the intermediary metabolism to inflammatory response, and even to the pathobiological evolution of critical diseases[47–49]. A detailed understanding of the heterogeneity of redox signaling and precisely deciphering their sophisticated correlations in pathogenesis demands the multiplex identification of various radical dynamics in complex living settings with high specificity and excellent tissue transparency. Therefore, development of unified and cutting-edge imaging modalities, which possess competent capability to dynamically map different radical variations in real time, and to spatiotemporally profile the redox-mediated pathophysiological implications at different stages, are still highly essential in clinics.

By taking advantages of the unique multiplexing emissions in NIR light-mediated UCNs and improved in vivo imaging depth in MSOT modality, herein, we created a promising strategy by technically anchoring the ROS-responsive HCy5 and RNS-responsive Cy7 fluorophores onto UCN nanocrystals with their maximum absorptions matching the upconverted emissions at 660 and 800 nm, respectively. Such effectively spectral overlap facilitated efficient energy transfer for optical imaging of RNS/ROS in one unified system. Importantly, the HCy5-Cy7 modification endowed UCNs with a broader chromogenic capability, which guaranteed their feasibility for simultaneous MSOT imaging of ROS/RNS with deeper penetration than traditional modalities through fluorescence or bioluminescence techniques.

Indeed, upon radical treatment, the ROS-responsive HCy5 on UCN surface regenerated the π-conjugation in Cy5, leading to an enhanced OA signal at 680 nm, but a decreased UCL signal at 660 nm. At the same time, the Cy7 moiety underwent structural degradation, resulting in a decreased OA change and recovery of UCL emission at 800 nm. Such significant OA and UCL changes mediated by radical oxidation allowed the rapid redox recognition with reasonable sensitivity down to a nanomolar range, thus providing an ideal nanoprobe for dynamic differentiation of ROS/RNS and further validation of their correlations with various inflammation stages in real time.

During MSOT imaging of radical stresses stimulated by LPS or APAP, the HCy5-Cy7-coated UCNs exhibited different signal response towards ROS/RNS reaction, in which the ROS production (e.g., by LPS) resulted in Cy5 regeneration on the particle surface, thus presenting a time-dependent signal enhancement at 680 nm, but little change at 800 nm concomitantly, while the nitrosative stress (e.g., by APAP) caused Cy7 degradation, indicating an obvious OA decrease at 800 nm and negligible change at 680 nm. Such opposite trend in two-channel OA alternations made UCN a remarkable nanoprobe for ratiometric monitoring of ROS/RNS distribution and variations in a highly precise manner.

We further examined the possibility of multiple radicals as intrinsic biomarkers to map out the undefined mechanisms of drug inactivation and closely track the dynamic inflammation processes upon diverse hepatotoxin administration. The extensive studies demonstrated that both APAP and INH treatment led to the initial enhancement of ROS along with more predominant RNS increasing subsequently in APAP or INH-treated mice. Compared to APAP, INH exhibited a similar CYP450 expression, suggesting that the radical production mainly occurred in phase I metabolism. However, more significant UGT consumption observed in INH metabolism implied the disparate metabolic

pathways between APAP and INH, and the glucuronidation-dominated pathway in phase II metabolism might be the main process to excrete INH from the body.

Interestingly, different from APAP and INH, live mice treated with THA, one FDA withdrawn anti-Alzheimer drug owing to its severe hepatotoxicity with the mechanisms under less elucidation, exhibited a major ROS but negligible RNS generation. Meanwhile, the THA metabolism resulted in more hepatic CYP450 expression and UGT consumption, as evident by the significant drug metabolism in phase I and II pathways. The pathological analysis through different inflammatory cytokines (e.g., IL-6 and IL-10) and liver injury indicators (e.g., ALT and AST) further demonstrated that THA would trigger more acute inflammation response and liver damage than those of APAP and INH, clearly suggesting the specific pathways of THA metabolism, which involved in phase I CYP450-mediated oxidation and the reactive metabolite (e.g., 1-hydroxytacrine, etc.) transformation, would be the potential reason for the acute hepatotoxicity[45]. These results demonstrated the reality of multiple redox species preceding the occurance of inflammation and histologically determined liver damage, thus enabling ROS/RNS as endogenous biomarkers to differentiate the dynamic drug metabolism and diseases pathophysiology at the early stages.

In summary, we present an innovative strategy by integration of unique NIR light-mediated upconverting nanoprobe with multispectral MSOT imaging for reversed ratiometric tracking of disparate oxidative and nitrosative stresses in vitro and in vivo. Such a promising strategy realizes the multiplex screening of diverse endogenous biomarkers and precise interrelation their dynamic correlations in complex living settings through one unified imaging modality, which is particularly meaningful to systematically exploit the underlying metabolism pathways and to orthogonally map out pathological implications towards the diseases at their different stages. This study not only facilitates better understanding of the pathophysiological roles of various redox species in living animals with non-invasive manner, but, more importantly, it also promoted the development of MSOT technology towards the exploration of undefined mechanism of potential pathogenesis in clinics, which may thus boost the pharmaceutical industry for high-throughput drug screening and pathological profiling of dynamic inflammation processing to benefit the healthcare communities in the future.

## Methods

**General**. Synthetic procedures and chemical characterizations of all the fluorophores and nanoplatforms are described in the Supplementary Methods.

**Endogenous radical species monitoring in live cells**. The murine macrophages RAW264.7 cell lines were from American Type Culture Collection (ATCC, cat. no. TIB-71) and checked for mycoplasma contamination, which was not listed by International Cell Line Authentication Committeeas misidentified cell lines. The cells were seeded in confocal dish overnight at a density of $1 \times 10^5$ cells mL Dulbecco's modified Eagle's medium. The excessive ROS (e.g., $O_2^{\bullet-}$) was activated by incubating cells with PMA (200 ng mL$^{-1}$) for 1 h, and RNS (e.g., ONOO$^-$) was achieved by stimulating with LPS (1 μg mL$^{-1}$) and INF-γ (50 ng mL$^{-1}$) for 4 h, followed by PMA (10 nM) treatment for 0.5 h. These radicals could be scavenged by pretreating with MnTBAP (100 μM) for $O_2^{\bullet-}$ and MEG (100 μM) for ONOO$^-$ at 1 h before stimuli addition. After refreshing the medium, the cells were incubated with UCNs (100 μg mL$^{-1}$) for 4 h, and the UCL imaging was performed in Nikon confocal microscopy using a continuous-wave 980 nm laser as excitation source (5 W cm$^{-2}$). The OA signals were collected by a commercial MSOT imaging system (iThera Medical, Germany) using a 128-element concave transducer array spanning a circular arc of 270° with the optimal excitation wavelength at 680–980 nm.

**Simultaneous screening of multiple radical dynamics in vivo**. All animal experimental procedures were performed in accordance with the protocols approved by the Institutional Animal Care and Use Committee of Soochow

University. Female Balb/c nude mice (~6–8 weeks old) were purchased from Shanghai Laboratories Animal Center in China. The mice were fasted overnight and i.p. injected with saline containing various drugs including LPS (20 mg kg$^{-1}$), APAP (300 mg kg$^{-1}$), INH (200 mg kg$^{-1}$), and THA (30 mg kg$^{-1}$), respectively, which could be pre-treated with NAC (200 mg kg$^{-1}$) as radical scavenger at 1 h before drug stimulation ($n = 5$). Fifteen minutes after drug treatment, the mice were tail vein injected with UCNs (5 mg mL$^{-1}$ in 100 μL saline) and anesthetized with 3% isoflurane for UCL imaging at 660 and 800 nm on the IVIS Lumina II imaging system with specific filters upon 980 nm excitation (10 W cm$^{-2}$). The in vivo MSOT imaging was further performed by whole-body screening along the long axis of mice (0.3 mm step distance, 10 repeat pulse per position) from 680 nm to 980 nm at designed time points after drug and UCN administration, and the averaged OA signal intensity in ROI region of liver was measured by the iThera MSOT imaging software.

**Histological, metabolic, and inflammatory analysis in vivo**. The mice were euthanized at 60 and 180 min upon UCN, APAP, INH, and THA treatment as described above. The organs including liver, heart, spleen, lung, and kidney were harvested and placed into 4% formalin solutions overnight for further H&E and immunohistochemical staining (4-HNE and 3-nitrotyrosine) by following the manufacturer's methods. All images were acquired using an Olympus IX53 inverted fluorescence microscope equipped with a Nuance (CRi Inc.) hyperspectral camera capable of bright field full-color imaging. Moreover, the livers were resected at designed time points after drug treatment and homogenized in ice-cold PBS for the assays of several hepatic biomarkers (e.g., CYP450, UGTs, IL-6, and IL-10) by enzyme-linked immunosorbent assay kit according to the standard protocols. The blood was also collected from the vena cava of live mice at different time points after drug treatment, and the serum was separated immediately to measure AST and ALT by following the manufacturer's procedures ($n = 5$).

**Statistical analysis**. Quantitative data are represented as mean ± standard deviation (SD). All of the measurements are taken from distinct samples, and the statistical significance are assessed by a Student's $t$-test (heteroscedastic, two-sided): *$p < 0.05$, **$p < 0.01$, ***$p < 0.001$).

## Data availability
The authors declare that all the data supporting the findings of this study are available from the authors on reasonable request.

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

## Acknowledgements

We thank Prof. Xiaogang Liu and Dr. Liangliang Liang for the luminescence lifetime tests. We also thank Prof. Edwin K.L. Yeow, Dr. Xiangyang Wu, and Zhizhong Chen for the helpful suggestions and discussions provided in data collection and analysis. This work was partially supported by NTU-AIT-MUV NAM/16001, Tier 1 RG5/18 (S), RG110/16 (S), (RG 35/15) NTU-JSPS JRP grant (M4082175.110), SSIJRI, and Merlion 2017 program (M408110000) awarded in Nanyang Technological University (NTU), National Natural Science Foundation of China (NSFC) (No. 51628201, 21874097, 81530057), National Key Research Program of China (2018YFA0208800), Collaborative Innovation Center of Radiation Medicine of Jiangsu Higher Education Institutions, and the Priority Academic Program Development of Jiangsu Higher Education Institutions (PAPD).

## Author contributions

B.X., X.A. and Z.W. conceived the idea and designed the experiments. M.G., Y.W., R.Z., J. L. and Y.Z. provided the devices for solutions and animal experiments, including MSOT and IVIS animal imaging systems. B.X., X.A., Z.W. and H.C. discussed the results and co-wrote the manuscript.

## Additional information

**Competing interests:** The authors declare no competing interests.

