## [Peer Review File · Nature Communications]

Reviewers' comments:

Reviewer #1 (Remarks to the Author):

Recommendation: Publish after minor revisions noted.

Comments:

The authors presented an innovative strategy by integration of unique NIR light-mediated upconverting nanoprobe with multispectral MSOT imaging for reversed ratiometric tracking of disparate oxidative and nitrosative stresses in vitro and in vivo. They provide precise profiling of the dynamic correlations and pathophysiological implications of multiplex biological mediators with deep penetration and highly programmed precision. Their work is interesting and could be published after making minor revisions according to the follow suggestions.

1. I noticed that the cyanine fluorophores HCy5 and Cy7 were loaded on the surface of PEG/PEI-UCNs by hydrophobic interaction. Are the loaded Cy7 and HCy5 molecules on the surface of UCNs stable in mouse serum?
2. Did the diameter change after loading Cy7 and HCy5 molecules on the surface of UCNs?
3. How about the stability of the loaded Cy7 and HCy5' s fluorescence upon incubation of the UCN in undiluted mouse serum?
4. Did the quantum yields of HCy5 and Cy7 decrease or increase after loading on the surface of UCNs?
5. The in-text graphic Figure 3e appears blurry.

Reviewer #3 (Remarks to the Author):

The submission by Xing et al. reports the development of a nanoprobe for simultaneous imaging of ROS and RNS in vivo. The probe design concept is pretty clever: by combining two ROS and RNS responsive cyanine dyes into upconversion nanoparticles (UCN), the probe provides two sets of signals: luminescence emission from UCNs, and optoacoustic signal from cyanine dyes, both of which show ROS/RNS dependent changes. A significant amount of work has been presented from the probe concept, in vitro characterization, cell validation to in vivo demonstration. While the overall study appears an extension of previously published work (ref #9), it has made valuable contributions on several aspects that are potentially of great interest to many in the nano and imaging community; for example, the clever coupling of absorption and luminescence emission among the dyes and UCNs; the ability of multiplex imaging of ROS and RNS with two independent signal channels. In my opinion, this work merits publication in Nature Communication. However, before publication, I would like to propose a revision on the following points.

There are a number of peaks involving the signal detection: the luminescence emission peaks of UCN, and the absorption peaks of HCy5 and Cy7 before and after ROS/RNS. It is my understanding that the luminescence emission peaks should be 660 nm and 800 nm, and upon ROS/RNS, 660 nm decreases due to the generation of Cy5 (from HCy5) that has an absorption peak at 640 nm, 800 nm peak increases because of the loss of Cy7 that has absorption peak at 750 nm. The paper seems to use 650 nm for both the luminescence emission peak at 660 nm and the 640 nm Cy5 absorption peak, and sometimes also refer the 660 nm peak as UCL 680 nm (Figure 2a). The author should check and refer to the maximum wavelength of the relevant peaks.

On the LRET: as Figure S3b shows, UCNs with the Cy7 encapsulation have little 800 nm emission. Was it due to the self-quenching of Cy7 dye? On the other hand, Figure S8b shows some Cy7 emission upon direct excitation at 770 nm. This would also apply to the ROS situation: it seems Cy5 does not emit in UCNs when excited at 980 nm. In addition, without the measurement of LRET efficiency, the authors cannot count on LRET as the sole mechanism and rule out the emission re-

absorbance pathway.

The authors may like to add a spectrum of luminescence and absorbance with the treatment of ROS/RNS in combination to Figure 2b and 2f. On Figure 2f: why RNS treatment increases 660 nm emission? Does RNS also break down HCy5? Figure 2e: very confusing with the combined data. I would suggest to convert it into two separate figures.

On the amount of HCy5 and Cy7 (8.9% and 12.2%): define % of what? Weight of the whole NPs?

Discussion on the nanoprobe uptake in liver and biodistribution, and confirmation of the probe in hepatocytes not just in Kupffer cells in liver would further strength the conclusions.

Reviewer #4 (Remarks to the Author):

Ai and colleagues report new probes that rely on upconverting nanoparticles to visualize oxidative and nitrosative stress in cells and in vivo. The approach is similar in principle to previous dual-labeled nanoparticles reported by Shuhendler et al., *Nat Biotech*, 2014, which were based on combined fluorescence-resonance energy transfer (FRET) and chemiluminescence resonance energy transfer (CRET). In the present work, the authors attempt to improve on past designs by extending the approach to optoacoustic (OA) imaging modalities that may enable monitoring deeper within tissue. Previous work has used OA imaging to visualize oxidative or nitrosative stimulation, but literature on dual ROS/RNS imaging using this approach is scant. Therefore, this present work may be of general interest. Several experiments may be performed to improve the publication however, which cut at the question: can this technology truly measure ROS and RNS simultaneously?

(1) Ratiometric imaging is emphasized for determining the balance of ROS:RNS, especially in Fig. 2. This raises several questions:

(1a) to what extent is the approach limited to examining relative changes in ROS:RNS (or even more limiting: examining either ROS or RNS)? Ideally, the authors should perform experiments with known combinations of ROS and RNS, and use imaging to infer the mixture.

The main disadvantage of this approach compared to Shuhendler lies in lack of orthogonality — ONOO decreases 680 and 800 signals. Therefore it is critical that we understand how to interpret the signal when combinations of ROS and RNS are anticipated. Perhaps quantitative mathematical deconvolution methods are required. This analysis needs to be done at every level — in vitro (with purified agents); in cells; and in vivo.

(1b) the combinations of cell stimuli show (at least naively) non-intuitive responses that are not really fleshed out or validated by the authors using alternative and well-known techniques (or cited literature). For instance PMA stimulates ROS but only in the absence of LPS/IFN γ ? How do we know that? Also - PMA alone stimulates ROS according to 3b-c, but the mitochondria are damaged similarly to the RNS stimulation condition? Are we sure there is no RNS being generated also?

The text is confusing probably has a typo: "The flow cytometry (FCM) results showed that the macrophage cells mainly presented green emission in the lower right quadrant upon stimulation with generators of RNS (~ 90 %, group 3) and ROS (~ 81 %, group 5) (Fig. 3e)."

But Group 3 was actually ROS generating?

Would be good to have alternative validation for the RNS/ROS signals observed in Fig 3.

(2) What are the dynamics of nanoparticle response to free-radical exposure?

(2a) What are the dynamics once the stimulus is removed? Is the signal stable? Especially with respect to ratio.

(3) How does prolonged imaging (of all modalities) impact the level and ratio in signals?

(4) How does tissue depth impact the ratio in signals? can this be calibrated? were the OA images calibrated / corrected for this? would be good to show phantom tissue images with / without correction for depth.

(5) APAP stimulates ROS (e.g., Shuhendler et al) but the authors' OA data here suggest otherwise (and it is confusing to interpret 4d and 4f together). The presumption is that the 680 signal is killed by RNS? So is this really dual detection?

(6) Protein induction of CYP450 by 30% in 7 minutes is surprisingly fast — please elaborate on how to interpret this finding.

(7) The authors use TUNEL as a marker of "DNA fragmentation associated with the ROS-induced cell damage". More selective markers of ROS-mediated (more than RNS-mediated) damage should be added, since many processes can lead to TUNEL-positivity. TUNEL is good to keep however as a marker of downstream cell response.

(8) Inline with the above comment, it would be good to have an ROS-signature treatment to compare with in Fig 6c-h (such as LPS, or early 60 min THA?). S18 has no time point labeled.

(9) Particle stability should be tested in media and ideally tissue homogenate.

(10) Longer term toxicity to UCN should be tested, especially since PEI is used which has known problems, especially branched PEI as used here. Nanomedicine (Lond). 2014 Feb; 9(2): 295–312.

Minor points:

What is shown in 2d that is not apparent in 2e?

Please better explain what the circles are besides figures 2e, 3b (presumably wells from a 96-well plate but where are the cells? what is the scale? etc); can't absolute values be used on all heat map images?

Groups 1-6 could be labeled embedded in the figure, it is a little inconvenient to find what the numbers correspond to in the caption.

Fig S8: the spectra should extend to 600nm as in Fig. S7.

INA and THA are capable of inducing oxidative or nitrosative stress (should be INH)

Where is control group in 6b?

Sample size, error bar definition, p-value, and statistical test used are missing in many panels.

Point-by-Point Response letter

The followings are our point-by-point response to the comments of reviewers and the changes made to the manuscript.

Reviewer #1 (Remarks to the Author):

Recommendation: Publish after minor revisions noted.

The authors presented an innovative strategy by integration of unique NIR light-mediated upconverting nanoprobe with multispectral MSOT imaging for reversed ratiometric tracking of disparate oxidative and nitrosative stresses in vitro and in vivo. They provide precise profiling of the dynamic correlations and pathophysiological implications of multiplex biological mediators with deep penetration and highly programmed precision. Their work is interesting and could be published after making minor revisions according to the follow suggestions.

Comment 1:

[1. I noticed that the cyanine fluorophores HCy5 and Cy7 were loaded on the surface of PEG/PEI-UCNs by hydrophobic interaction. Are the loaded Cy7 and HCy5 molecules on the surface of UCNs stable in mouse serum?]

Response:

We thank the reviewer's time and effort in reviewing our manuscript. As suggested by the reviewer, we incubated Cy7 and HCy5 loaded UCNs with undiluted mouse serum for different time duration. And the stabilities of the loaded Cy7 and HCy5 molecules on UCNs surface were evaluated based on their completely different absorption spectra, as shown in Fig. R1a. After 24 h incubation, the maximum absorbance peak of Cy7 at 780 nm and HCy5 at 385 nm still remain ~ 90% and 85%, respectively (Fig. R1b), which are comparable with the relevant stability tests reported previously (e.g. *ACS Nano*, 2016, 10, 10049; *Angew. Chem. Int. Ed.*, 2013, 52, 10325; etc). These results clearly indicated that the Cy7/HCy5-loaded UCNs can act as a stable nanoplatform in physiological environment for further *in vivo* studies. We have included these information in our revised manuscript and supplementary information.

Fig. R1. (a) Absorbance spectra of Cy7, HCy5 and UCNs; (b) Stability tests of loaded Cy7 and HCy5 molecules on UCNs surface at 780 nm and 385 nm, respectively in mouse serum upon different time incubation.

Comment 2:

[2. Did the diameter change after loading Cy7 and HCy5 molecules on the surface of UCNs?]

Response:

According to the reviewer's comments, we examined the size distribution of UCNs nanostructures before and after Cy7 and HCy5 molecules loading on particles surface (Cy7/HCy5-UCNs) by TEM and DLS analysis. As shown in Fig. R2, both the TEM images of UCNs particles with or without fluorophores loading indicated a well-distributed and uniform morphology (~ 20 nm). The DLS analysis presented a slightly change of hydrodynamic diameter (95 ± 17 nm and 100 ± 27 nm). These results showed that there is no obvious diameter change after embedding Cy7 and HCy5 molecules in the polymer layer on the surface of UCNs *via* hydrophobic interaction, which are consistent with the related data reported before (*e.g.*, *Theranostics*, 2018, 8, 1435; *Biomaterials*, 2015, 54, 34; *etc*). We have included these new data in our revised manuscript and supplementary information.

Fig. R2. TEM and DLS analysis of UCNs alone (a) and Cy7/HCy5-loaded UCNs (b). Scale bar: 100 nm.

Comment 3:

[3. How about the stability of the loaded Cy7 and HCy5's fluorescence upon incubation of the UCN in undiluted mouse serum?]

Response:

We appreciate the reviewer for bringing out this point regarding the fluorescence stability of loaded-Cy7 and HCy5 molecules on UCN surface in undiluted mouse serum. Basically, the rationale of our design is to establish a unique nanoprobe by integration two ROS/RNS responsive fluorophores (Cy7 and HCy5) on UCNs surface for simultaneous visualization of the diversely endogenous redox biomarkers at dual wavelength channels. Typically, the loaded Cy7 fluorophores present strong fluorescence at 800 nm on UCNs surface, which will gradually decrease upon the treatment of RNS (*e.g.*, ONOO⁻) due to the irreversible oxidative degradation of Cy7 structure. While, the loaded HCy5 molecules display negligible fluorescence at the beginning, which only demonstrate significant fluorescence increase at 660 nm after responding to ROS (*e.g.*, O₂^{•-}) treatment owing to the oxidation-mediated HCy5 regeneration to Cy5 structures.

Therefore, by following the reviewer's suggestion, we investigated the stabilities of the loaded Cy7's fluorescence on UCNs surface upon incubation in undiluted mouse serum. And the potential fluorescence change at 800 nm was monitored at different time duration. As shown in Fig. R3, in the absence of RNS treatment, there was no obvious fluorescence change observed for Cy7 (at 800 nm) after 24 h incubation, suggesting that the loaded Cy7 fluorophores on the surface of UCNs presented great stabilities in physiological environment for further imaging studies *in vivo*.

Fig. R3. Fluorescence stabilities of the loaded Cy7 molecules on UCNs surface at 800 nm (E_x : 770 nm) in undiluted mouse serum at different time duration.

Comment 4:

[4. Did the quantum yields of HCy5 and Cy7 decrease or increase after loading on the surface of UCNs?]

Response:

We thank the reviewer's time and effort in reviewing our manuscript. By following the reviewer's comments, we determined the quantum yield (QY) of Cy7 and HCy5 before and after loading on the UCNs nanopatform. For Cy7, before loading, the QY of Cy7 fluorophore was $\sim 5.10 \pm 0.61$ %, which is similar with the value reported previously (e.g., *small*, 2016, 12, 1968; *J. Phys. Chem. B.*, 2006, 110, 16491; *ACS Appl. Mater. Interfaces*, 2016, 8, 29899, etc). After loading on UCNs surface, the QY of Cy7 fluorophore became $\sim 3.89 \pm 0.34$ %. The decreased QY before and after loading to UCNs indicated the possibility of the energy transfer between Cy7 and UCNs. As expected, there was no obvious QY value observed for HCy5 before and after loading on UCNs surface. This is consistent with our rational design mentioned earlier: there was negligible fluorescence detected for HCy5 molecule itself, which only presents strong fluorescence at 660 nm after reaction with ROS radicals. These results have been included in our revised manuscript and supplementary information.

Comment 5:

[5. The in-text graphic Figure 3e appears blurry.]

Response:

We appreciate the reviewer's suggestion. We have revised the figures in our manuscript by adjusting the resolution of graphic text in a more distinct manner.

Reviewer #3 (Remarks to the Author):

The submission by Xing et al. reports the development of a nanoprobe for simultaneous imaging of ROS and RNS in vivo. The probe design concept is pretty clever: by combining two ROS and RNS responsive cyanine dyes into upconversion nanoparticles (UCN), the probe provides two sets of signals: luminescence emission from UCNs, and optoacoustic signal from cyanine dyes, both of which show ROS/RNS dependent changes. A significant amount of work has been presented from the probe concept, in vitro characterization, cell validation to in vivo demonstration. While the overall study appears an extension of previously published work (ref #9), it has made valuable contributions on several aspects that are potentially of great interest to many in the nano and imaging community; for example, the clever coupling of absorption and luminescence emission among the dyes and UCNs; the ability of multiplex imaging of ROS and RNS with two independent signal channels. In my opinion, this work merits publication in Nature Communication. However, before publication, I would like to propose a revision on the following points.

Comment 1:

[1. There are a number of peaks involving the signal detection: the luminescence emission peaks of UCN, and the absorption peaks of HCy5 and Cy7 before and after ROS/RNS. It is my understanding that the luminescence emission peaks should be 660 nm and 800 nm, and upon ROS/RNS, 660 nm decreases due to the generation of Cy5 (from HCy5) that has an absorption peak at 640 nm, 800 nm peak increases because of the loss of Cy7 that has absorption peak at 750 nm. The paper seems to use 650 nm for both the luminescence emission peak at 660 nm and the 640 nm Cy5 absorption peak, and sometimes also refer the 660 nm peak as UCL 680 nm (Figure 2a). The author should check and refer to the maximum wavelength of the relevant peaks.]

Response:

We appreciate the reviewer's constructive comments on our manuscript. By following the reviewer's valuable suggestions, we check through the manuscript and refer to the maximum wavelength of the relevant peaks for effective signal detection. For example, upon RNS (e.g., ONOO⁻) and ROS (e.g., O₂^{•-}) treatment, the maximum absorbance changes of the loaded HCy5 and Cy7 molecules on UCN surface were determined at 640 nm and 780 nm. The maximal upconversion luminescence (UCL) emissions at 660 nm and 800 nm from UCNs were used to present individual signal changes when upon the stimulation of UCNs with ROS (e.g., O₂^{•-}) and RNS (e.g., ONOO⁻), respectively. For optoacoustic (OA) analysis, since the available NIR light excitation in most commercially existing MSOT equipment ranges from 680 – 980 nm. Therefore, the maximum OA signal changes of UCNs nanoprobes upon ROS and RNS treatment were demonstrated at 680 nm and 800 nm respectively. We have updated these details in our revised manuscript and supplementary information.

Comment 2:

[2. On the LRET: as Figure S3b shows, UCNs with the Cy7 encapsulation have little 800 nm emission. Was it due to the self-quenching of Cy7 dye? On the other hand, Figure S8b shows some Cy7 emission upon direct excitation at 770 nm. This would also apply to the ROS situation: it seems Cy5 does not emit in UCNs when excited at 980 nm. In addition, without the measurement of LRET efficiency, the authors cannot count on LRET as the sole mechanism and rule out the emission re-absorbance pathway.]

Response:

We appreciate the reviewer for such valuable comments to improve our studies. By following the reviewers' suggestions, we performed the detailed spectroscopic analysis to explore the potential pathways for the luminescent change upon surface loading with dye molecules.

First, we measured the fluorescence emissions of free Cy7 fluorophore and the same concentration of Cy7 molecules loaded on UCNs surface upon their excitation at 770 nm. As shown in Fig. R4, compared to free Cy7, the emission of Cy7 after loading on UCNs surface presented obvious decrease of emission at 800 nm. Moreover, we also measured the potential lifetime change of Cy7 molecules before and after loading on UCNs nanoparticles. As indicated in Fig. R4 (inset), the lifetime of Cy7 loaded on particle surface decreased from 1.18 ns to 0.48 ns. Such lifetime change was consistent with the related studies reported previously (e.g., *small*, 2016, 12, 1968; *ACS Nano*, 2017, 11, 11264; *J. Am. Chem. Soc.*, 2012, 134, 4545, etc). These measurements suggested the possibility of the self-quenching effect of Cy7 fluorophores on UCNs surface. Moreover, as pointed out by reviewer, the similar case can be also applied for ROS responsive HCy5. In our system, the loaded HCy5 molecules show negligible fluorescence, which demonstrate significant fluorescence increase at 660 nm only after treatment with ROS (e.g., O₂^{•-}). Therefore, HCy5 would not emit in UCNs when excited at 980 nm.

Fig. R4. The fluorescence of free Cy7 and the same concentration (2.0 μ M) of Cy7 molecules loaded on UCNs probe (E_x : 770 nm). Inset: luminescence decay curves of Cy7 emission at 800 nm before and after encapsulated on UCNs surface.

Moreover, by following the reviewer's suggestion, we also measured the LRET efficiency based on the methods established previously (e.g., *Chem. Asian J.*, 2018, 13, 614; *Adv. Mater.*, 2018, 30, 1801726; *Anal. Chem.*, 2017, 89, 4868, etc).

Typically, the LRET efficiency (E) can be calculated based on the following formula:

$$E = 1 - \tau_{DA}/\tau_D$$

Where τ_{DA} and τ_D represent the lifetime of the donor (UCNs) with or without loading of the acceptor (fluorophores), respectively.

As shown in Fig. R5, upon 980 nm excitation, the lifetime of UCNs at 800 nm decreases from 251 μ s to 162 μ s after Cy7 encapsulation. Accordingly, the LRET efficiency of Cy7 fluorophores on UCNs surface were calculated as $\sim 35.4\%$, which are comparable with the values reported before (e.g., *J. Am. Chem. Soc.*, 2015, 137, 3421; *Angew. Chem. Int. Ed.*, 2017, 56, 4165, etc). Compared to the significant UCNs luminescence decrease at 800 nm after encapsulation of Cy7 in Figure S3b, these LRET efficiency results suggested that LRET may not the sole mechanism of energy transfer process from UCNs to the encapsulated fluorophores. The possibility of the emissions from the re-absorbance pathway could not be completely ruled out (e.g., *J. Am. Chem. Soc.*, 2016, 138, 15972; *J. Phys. Chem. C*, 2011, 115, 17736; *Nanoscale*, 2018, 10, 250, etc). We appreciate the reviewer for such valuable suggestions here. And we have included this information in our revised manuscript and supplementary materials.

Fig. R5. The UCL luminescence decay curves of the UCNs nanoparticles at 800 nm with (red) or without (gray) surface encapsulation of Cy7 (E_x : 980 nm).

Comment 3:

[3. The authors may like to add a spectrum of luminescence and absorbance with the treatment of ROS/RNS in combination to Figure 2b and 2f. On Figure 2f: why RNS treatment increases 660 nm emission? Does RNS also break down HCy5? Figure 2e: very confusing with the combined data. I would suggest to convert it into two separate figures.]

Response:

We appreciate the reviewer's constructive comments here. By following the reviewer's suggestions, we have included a spectrum of absorbance and luminescence of UCNs nanoprobe with the treatment of combined ROS/RNS. As expected, the stimulation of ROS (e.g., $O_2^{\cdot-}$, 20 μM) and RNS (e.g., ONOO $^-$, 20 μM) in combination with UCNs probe could simultaneously cause the obvious absorbance increase at 640 nm and decrease at 780 nm (in Fig. R6a), mostly attributed to the ROS-induced Cy5 regeneration (from HCy5) and RNS-mediated Cy7 degradation, respectively. Such opposite radical oxidation-responsive spectral changes provided unique advantages of UCNs nanoprobe as MSOT imaging contrast agents to selectively differentiate ROS and RNS stresses in living settings.

Fig. R6. Absorbance (a) and UCL (b) spectra of UCNs nanoprobe upon ROS ($O_2^{\cdot-}$, 100 μM), RNS (ONOO $^-$, 100 μM) and their combination (ROS: 20 μM , RNS: 20 μM) treatment (E_x : 980 nm).

Similarly, we also determined the upconversion luminescence (UCL) spectra of UCNs nanoprobe upon the combined ROS/RNS treatment. As shown in Fig. R6b, RNS (e.g., ONOO $^-$) treatment led to an obvious UCL signal recovery at 800 nm, mainly due to the degradation of Cy7 to reduce the energy transfer between UCNs to Cy7. Such RNS responsive structural degradation could lead to a little decrease of the weak leading absorbance of Cy7 at 660 nm which thus resulted in a slightly enhanced UCL emission at this wavelength. Our spectral analysis confirmed that the RNS (e.g., ONOO $^-$) treatment will not affect HCy5 molecule, which could be activated only by ROS reaction and contributed to a decreased UCL emission at 660 nm, opposite from the emission change caused by RNS-responsive Cy7. Indeed, as shown in Fig. R6b, there was an obvious UCL emission decrease

observed at 660 nm upon ROS stimulation, suggesting the ROS-mediated Cy5 reproduction (from HCy5) to enhance the energy transfer process from UCNs to the regenerated Cy5 structures. While, ROS treatment would not lead to the obvious UCL emission change at 800 nm, indicating the specificity of Cy7 for RNS recognition that will not be interfered by ROS radicals. These results demonstrated that HCy5/Cy7 loaded UCNs nanoprobe can work as effective MSOT contrast agents for reversed ratiometric imaging of ROS and RNS stresses in complex living conditions.

Moreover, according to the reviewer's suggestions, we converted the Fig. 2e to two separate figures (Fig. 2e and 2f in the revised manuscript) to help the readers easily understand the proposed studies. All the relevant information has been included in our revised manuscript already.

Comment 4:

[4. On the amount of HCy5 and Cy7 (8.9% and 12.2%): define % of what? Weight of the whole NPs?]

Response:

Thanks for the reviewer's comments. In our studies, the loading amounts of Cy7 and HCy5 molecules were determined based on their percentages of weight on the whole UCNs nanoprobe by following the established strategies (e.g., *Biomaterials*, 2011, 32, 1110; *J. Mater. Chem. B*, 2016, 4, 4884, etc). Typically, the loaded Cy7 and HCy5 molecules were completely extracted by organic solvents (e.g., DMSO) for several times according to the reported methods previously (e.g., *Theranostics*, 2018, 8, 1435; *Angew. Chem. Int. Ed.*, 2013, 52, 10325; *ACS Nano*, 2016, 10, 10049, etc), and their concentration was further analyzed by utilizing high performance liquid chromatography (HPLC) based on their specific maximum absorbance at 780 nm and 385 nm respectively (Fig. R7). The final loading amounts of HCy5 and Cy7 were determined based on their standard curves respectively. We have included all of these results in our revised manuscript and supplementary information.

Fig. R7. Quantification of Cy7 and HCy5 on UCNs nanoprobe. (a) The standard curve for Cy7 based on its absorbance at 780 nm. (b) The standard curve for HCy5 based on its absorbance at 385 nm. HPLC was applied to quantify the amounts of fluorophores on UCNs.

Comment 5:

[5. Discussion on the nanoprobe uptake in liver and biodistribution, and confirmation of the probe in hepatocytes not just in Kupffer cells in liver would further strength the conclusions.]

Response:

We greatly appreciate the reviewer's valuable suggestion to strengthen our studies. By taking the reviewers' suggestions, we carried out the histological studies systematically. The uptake and biodistribution of UCNs nanoprobe were carefully investigated in the frozen sections of the mouse liver tissues by monitoring of the upconverted luminescence (UCL) (with the NIR light excitation at 980 nm). As shown in Fig. R8, after tail-vein injection of UCNs probe for 2 hours, the significant

green UCL emission at 540 nm was observed in the whole liver tissues sections. The specific cell staining recognized by nuclear morphologies confirmed the distribution of UCNs nanoprobe in both Kupffer and hepatocytes cells (Fig. R8), which are consistent with those reported before (*e.g.*, *J. Control. Release.*, 2016, 240, 332; *ACS Nano.*, 2017, 11, 2428; *Hepatobiliary Surg. Nutr.*, 2014, 3, 364, *etc*). These results further strength the conclusions that our nanoprobe could effectively monitor the dynamic radical variations in liver with the pathological progression in living animals. We have included this information in our revised manuscript and supplementary materials.

Fig. R8. Fluorescence analysis of the UCNs nanoprobe uptake and biodistribution in mouse liver tissues. Blue: Hoechst 33342 (E_x : 405 nm, E_m : 460/50 nm); Green: UCL540 (E_x : 980 nm, E_m : 540/50 nm); K: Kupffer cells (red); H: hepatocytes (white). Scale bar: 20 μm .

Reviewer #4 (Remarks to the Author):

Ai and colleagues report new probes that rely on upconverting nanoparticles to visualize oxidative and nitrosative stress in cells and in vivo. The approach is similar in principle to previous dual-labeled nanoparticles reported by Shuhendler et al., Nat Biotech, 2014, which were based on combined fluorescence-resonance energy transfer (FRET) and chemiluminescence resonance energy transfer (CRET). In the present work, the authors attempt to improve on past designs by extending the approach to optoacoustic (OA) imaging modalities that may enable monitoring deeper within tissue. Previous work has used OA imaging to visualize oxidative or nitrosative stimulation, but literature on dual ROS/RNS imaging using this approach is scant. Therefore, this present work may be of general interest. Several experiments may be performed to improve the publication however, which cut at the question: can this technology truly measure ROS and RNS simultaneously?

Comment 1:

[(1) Ratiometric imaging is emphasized for determining the balance of ROS:RNS, especially in Fig. 2. This raises several questions:

(1a) to what extent is the approach limited to examining relative changes in ROS:RNS (or even more limiting: examining either ROS or RNS)? Ideally, the authors should perform experiments with known combinations of ROS and RNS, and use imaging to infer the mixture.

The main disadvantage of this approach compared to Shuhendler lies in lack of orthogonality — ONOO decreases 680 and 800 signals. Therefore it is critical that we understand how to interpret the signal when combinations of ROS and RNS are anticipated. Perhaps quantitative mathematical deconvolution methods are required. This analysis needs to be done at every level — in vitro (with purified agents); in cells; and in vivo.]

Response:

We sincerely appreciate the reviewer for such valuable comments and appraisal on our manuscript. Here, the reviewer concerned the approach to examining relative changes in ROS:RNS and suggested the experiments with the combination of ROS and RNS. Therefore, by following such valuable suggestion, we performed the spectral analysis of UCNs nanoprobe with the treatment of combined ROS/RNS. As shown in Fig. R9a, the combined stimulation of UCNs nanoprobe with ROS (e.g., $O_2^{\cdot-}$, 20 μ M) and RNS (e.g., ONOO $^-$, 20 μ M) led to the absorbance increase at 640 nm and signal decrease at 780 nm. The reverse signal changes were consistent with the UCNs treated with individual ROS or RNS respectively, mainly due to the ROS-induced Cy5 regeneration (from HCy5) and RNS-mediated Cy7 degradation. Such opposite spectral changes caused by different radicals enabled UCNs as an ideal probe for ratiometric OA imaging of the disparate ROS/RNS stress in living system.

More importantly, we greatly appreciate the reviewer's useful suggestions to interpret the signal based on the quantitative mathematical deconvolution methods. By following up such good points, we performed the detailed OA signals analysis by utilizing the mathematical deconvolution technique in all levels including in vitro buffers, in cells and in living animals.

First, we determined absorbance and OA spectrum of UCNs with the combined treatment of ROS/RNS (Fig. R9a and b).

Fig. R9. The absorption (a) and optoacoustic (b) spectra of UCN probe (1 mg mL^{-1}) in the absence and presence of both ROS ($\text{O}_2^{\cdot-}$, $20 \text{ }\mu\text{M}$) and RNS (ONOO^- , $20 \text{ }\mu\text{M}$) under the mixture condition.

Then, we inferred the mixture of ROS and RNS by the quantitative mathematical deconvolution method as reported previously (e.g., *Chem. Soc. Rev.*, 2000, 29, 217; *Angew. Chem. Int. Ed.*, 2016, 55, 11770; *Biomed. Opt. Express.*, 2016, 7, 369, etc).

Typically, the quantitative analysis of ROS and RNS in the mixture could be achieved based on the following established procedures:

In general, the spectral deconvoluted OA signals at 680 nm and 800 nm are directly proportional to the concentration of ROS-regenerated Cy5 (from HCy5) and RNS-responsive Cy7 molecules on UCNs surface, which could be denoted as the sum of each components in the mixture (or a 2×2 matrix form):

$$\begin{cases} \text{OA}_{680} = \varepsilon_a \cdot C_1 + \varepsilon_b \cdot C_2 \\ \text{OA}_{800} = \varepsilon_c \cdot C_1 + \varepsilon_d \cdot C_2 \end{cases} \quad \text{or} \quad \begin{pmatrix} \text{OA}_{680} \\ \text{OA}_{800} \end{pmatrix} = \begin{pmatrix} \varepsilon_a & \varepsilon_b \\ \varepsilon_c & \varepsilon_d \end{pmatrix} \begin{pmatrix} C_1 \\ C_2 \end{pmatrix}$$

where ε_a and ε_b are the proportionality constants of OA signals at 680 nm for the regenerated Cy5 (from HCy5); ε_c and ε_d are the proportionality constants of OA signals at 800 nm for Cy7 respectively; C_1 and C_2 indicate the concentration of fluorophores accordingly. Based on the process of the mathematical deconvolution, the OA spectra upon the combined treatment of ROS-RNS could be specifically separated at 680 nm and 800 nm, respectively (Fig. R9b).

We tested the OA spectra of purified agents (e.g. Cy7 and Cy5) with known concentrations, and obtained the values of these proportionality constants based on the slopes of their standard curves. Thus, the mathematical deconvolution could differentiate the response of ROS and RNS for their contribution to OA signals changes at different wavelengths.

Furthermore, similar mathematical deconvolution was also applied for MSOT imaging analysis in RAW264.7 macrophage cells. As shown in the Fig. R10, the deconvoluted OA signal presented an obvious enhancement at 680 nm but negligible change at 800 nm in group 3, suggesting the specific generation of ROS (e.g., $\text{O}_2^{\cdot-}$) upon PMA stimulation. Whereas, upon the treatment of LPS/IFN- γ /PMA, the deconvoluted OA signal displayed an obvious attenuation at 800 nm but less change at 680 nm (group 5), indicating the extensive RNS (e.g., ONOO^-) production in macrophage cells. These results could be further confirmed by the confocal microscopy based on the alternative standard fluorimetric intracellular peroxynitrite (green) and superoxide (red) assay tracker in Fig. R12 (as discussed in the response of next comment), clearly suggesting that the mathematical deconvolution will be a very powerful strategy to effectively differentiate the balance of ROS and RNS in living cells.

Fig. R10. Deconvoluted OA signals at 680 nm (a) and 800 nm (b) of macrophage cells upon PMA and LPS/INF- γ /PMA stimulation for ROS ($O_2^{\bullet-}$) and ROS ($ONOO^-$) generation in the absence and presence of $O_2^{\bullet-}$ inhibitor (e.g., MnTBAP) and $ONOO^-$ inhibitor (e.g., MEG), respectively.

More significantly, as suggested by reviewer, we also performed the mathematical deconvolution processes in living mice for dynamic profiling of the ROS and RNS variations upon the stimulation with different hepatotoxic drugs. As shown in Fig. R11, the dynamic variations of ROS and RNS over the imaging process could be clearly indicated by the deconvoluted OA signals at 680 nm and 800 nm respectively, further confirming that the generation of ROS and RNS in living system could be differentiated by our proposed UCNs nanoprobe through the MSOT imaging under mathematical deconvolution methods.

Fig. R11. Dynamic profiling of deconvoluted OA signal changes in hepatic inflammation models by UCNs at 680 nm (a, c) and 800 nm (b, d) upon LPS, APAP, INH, THA and their reactive metabolites scavenger (NAC) treatment ($n = 5$).

Finally, by following the reviewer's suggestions, we have proceeded the mathematical deconvolution analysis at every level — in buffers, in cells and in living mice. All these experimental results clearly prove the great potential of UCNs as a promising nanoprobe for the dynamic screening of oxidative and nitrosative stresses under dual different optoacoustic channels. These studies have been included into our revised manuscript and supplementary information. We appreciate the reviewer for such a great advice.

[(1b) the combinations of cell stimuli show (at least naively) non-intuitive responses that are not really fleshed out or validated by the authors using alternative and well-known techniques (or cited literature). For instance PMA stimulates ROS but only in the absence of LPS/IFNg? How do we

know that? Also - PMA alone stimulates ROS according to 3b-c, but the mitochondria are damaged similarly to the RNS stimulation condition? Are we sure there is no RNS being generated also?]

Response:

We greatly appreciate the reviewer’s kind suggestions to strengthen our cellular studies by using alternative techniques as additional supports. By following such valuable suggestions, we further confirmed the specific generation of ROS and RNS in macrophage cells by confocal microscopy through the alternative standard fluorimetric intracellular peroxynitrite (green) and superoxide (red) assay tracker (both are well-known assays for radical identification) (e.g., *Hepatology*, 2008, 47, 1248; *Free Radic. Biol. Med.*, 2005, 38, 286, etc). As shown in Fig. R12, the bright red fluorescence of superoxide indicator was easily visualized in RAW264.7 macrophage cells by pretreating with phorbol 12-myristate 13-acetate (PMA) (~ 300 nM) for 1 h, demonstrating the excessive activation of ROS (e.g., $O_2^{\cdot-}$) in living cells as reported previously (e.g., *J. Am. Chem. Soc.*, 2015, 137, 6837; *J. Am. Chem. Soc.*, 2013, 135, 14956; *Angew. Chem. Int. Ed.*, 2009, 48, 299, etc). As control, no significant signal could be determined in the cells without incubation of PMA or in the cells pre-treated with superoxide scavenger (e.g., MnTBAP), suggesting the specific generation of superoxide ($O_2^{\cdot-}$) species in macrophage cells. In addition, the bright green fluorescence of peroxynitrite indicator was also observed in cells upon stimulating with lipopolysaccharide (LPS, $1 \mu\text{g mL}^{-1}$) and interferon- γ (INF- γ , 50 ng mL^{-1}) for 4 h followed by PMA (10 nM) treatment for 0.5 h, indicating the excessive induction of RNS (e.g., ONOO $^-$) in macrophage cells (e.g., *J. Am. Chem. Soc.*, 2016, 138, 10778; *J. Am. Chem. Soc.*, 2010, 132, 2795; *Nat. Nanotechnol.*, 2014, 9, 233, etc). There were negligible signals observed in control studies when the cells were pre-treated with mercaptoethyl guanidine (MEG, $100 \mu\text{M}$) as ONOO $^-$ scavenger for 1 h. These imaging results further proved that the specific generation of ROS (e.g., $O_2^{\cdot-}$) and RNS (e.g., ONOO $^-$) could be achieved in macrophage cells upon the stimulations by PMA and LPS/INF- γ /PMA respectively. We also included these results in the revised supporting information.

Fig. R12. Scheme and confocal imaging of ROS/RNS generation in RAW264.7 cells upon the treatment of PMA and LPS/INF- γ /PMA followed by MnTBAP and MEG as specific scavenger for superoxide ($O_2^{\cdot-}$) peroxynitrite (ONOO $^-$) species, respectively. Blue: Hoechst 33342 (E_x : 405 nm, E_m : 460/50 nm), green: peroxynitrite indicator (E_x : 480 nm, E_m : 520/50 nm), red: superoxide indicator (E_x : 565 nm, E_m : 590/40 nm). Scale bar: 50 μm .

[The text is confusing probably has a typo: “The flow cytometry (FCM) results showed that the macrophage cells mainly presented green emission in the lower right quadrant upon stimulation with generators of RNS (~ 90 %, group 3) and ROS (~ 81 %, group 5) (Fig. 3e).”]

Response:

We apologize for the confusion caused by the typo here. We have rephrased the description in our revised manuscript: “*The flow cytometry (FCM) results showed that the macrophage cells population mainly located at the lower right quadrant upon stimulation with generators of ROS (~ 90 % in group 3) and RNS (~ 81 % in group 5) respectively (Fig. 3e).*”

[But Group 3 was actually ROS generating? Would be good to have alternative validation for the RNS/ROS signals observed in Fig 3.]

Response:

In group 3 of Fig. 3, the macrophage cells were pretreated by phorbol 12-myristate 13-acetate (PMA) (~ 300 nM) for 1 h to generate excessive ROS (e.g., O₂[•]). By following the reviewer’s suggestion, we have utilized the alternative ROS/RNS trackers (e.g., the standard fluorimetric intracellular peroxynitrite (green) and superoxide (red) assay tracker) to further re-validate the ROS/RNS signals (Fig. R12, as discussed earlier).

These results confirmed that the specific generation of ROS/RNS could be achieved in macrophage cells upon the combinations of cell stimuli treatment at different conditions. We have included these data in the revised manuscript and supporting information.

Comment 2:

[(2) What are the dynamics of nanoparticle response to free-radical exposure?]

[(2a) What are the dynamics once the stimulus is removed? Is the signal stable? Especially with respect to ratio.]

Response:

We appreciate the reviewer for bringing out this comment regarding the dynamics of nanoparticle response to free-radical stimulation. According to the reviewer’s suggestions, we have determined the dynamic optoacoustic (OA) signal changes of UCNs nanoprobe upon ROS and RNS treatment at different time duration. As shown in Fig. R13a, under ROS (e.g., O₂[•]) stimulation, the OA signal of UCNs probe at 680 nm presented a rapid enhancement at the initial 10 min, and the signal kept stable within the following 180 min. Similarly, RNS (e.g., ONOO⁻) treatment resulted in an obvious OA signal decrease at 800 nm at the initial 10 min, and there was no significant signal change observed within the next few hours (~ 3hr) (Fig. R13b). These results demonstrated the rapid response and great stability of OA signals upon the UCNs stimulation with ROS or RNS treatment.

Furthermore, as suggested, we also performed the dynamic studies when the ROS/RNS stimuli were removed by addition of free radicals scavenger (e.g., ascorbic acid, 100 μM) (e.g., *Scientific Report, 2013, 3, 2146; ACS Nano, 2012, 6, 10632;*). As shown in Fig. R13a-b, upon the removal of ROS or RNS by adding of scavenger, the amounts of OA signal decreased significantly at 680 nm or 800 nm, and the decreased OA signals maintained stable with the consistent ratiometric values for ~ 3 hrs. (Fig. R13c). These results suggested the reasonable dynamic stability of our proposed UCNs probe for their response to multiple redox species stimulation *in vitro* and *in vivo*.

Fig. R13. Dynamic analysis of UCNs nanoprobe for the response of radical species stimulation. OA signals at 680 nm (a) and 800 nm (b) upon treatment with ROS (O_2^- , 100 μ M), RNS ($ONOO^-$, 100 μ M) or scavenger (e.g. ascorbic acid, 100 μ M) respectively. (c) The ratiometric signal of UCNs probe after their reaction with different radicals.

Comment 3:

[(3) How does prolonged imaging (of all modalities) impact the level and ratio in signals?]

Response:

We appreciate the reviewer’s valuable comment on our manuscript. In our study, the rational design is to establish a unique MSOT nanoprobe to simultaneously image the diversely endogenous ROS/RNS biomarkers at two different optical channels, and to precisely validate their dynamics for redox-mediated pathophysiological procession. Technically, we treated the living mice with different drugs and systematically examined dose-dependent ROS/RNS activity as well as drug-induced inflammation in the liver within minutes of drug challenge. Considering the potential clearance of UCNs nanoprobe along with prolonged time duration (as shown in Fig. R14, the OA and UCL imaging results suggested by the reviewer), we proceeded ROS/RNS imaging and pathophysiological experiments under short period of time (within ~ 3 h) to make sure the signals collected are stable and reliable.

Fig. R14. Time-resolved variations of OA signals at 680 nm and 800 nm (a, b), and UCL signals at 660 nm and 800 nm (c, d) respectively in mouse liver upon UCNs nanoprobe administration ($n = 5$).

Comment 4:

[(4) How does tissue depth impact the ratio in signals? can this be calibrated? were the OA images calibrated / corrected for this? would be good to show phantom tissue images with / without correction for depth.]

Response:

We thank the reviewer for such professional comments. As an amazing imaging modality, MSOT can supply reliable anatomy information for effective theranostics *in vivo* and in preclinical testing. During the imaging process, the tissue depth could be a factor to potentially affect the signal collection, just as the reviewer pointed out. Therefore, as proof-of-concept suggested by the reviewer, we determined the OA signals of UCNs nanoprobe at 680 nm and 800 nm with the pork adipose tissues at different thickness and proceeded the calibration accordingly.

As shown in Fig. R15a-b, the OA signals at 680 nm displayed negligible changes when the thickness was less than 5 mm and gradually decreased along with the increment of tissue thickness. Similarly, the OA images at 800 nm also presented minimum variations within the tissue thickness less than 8 mm, which underwent continuous signal decrease with the depth of the tissue (up to 25 mm). Here, we calculated the ratiometric OA signals ($(\Delta OA_{680} + \Delta OA_{800}) / OA_{800}$) within different tissue thickness. As shown in Fig. 15c, the ratiometric values presented the reliable stability when the thickness was ~ 8 mm, which was sufficient for the determination of radicals-induced signal changes in mouse liver as we indicated in the main manuscript (e.g., *Chem. Sci.*, 2017, 8, 7025; *Nat. Commun.*, 2018, 9, 3983, etc).

For the tissue depth more than 8 mm, the ratiometric signals gradually decreased along with the increment of tissues thickness. Therefore, by following the reviewer's suggestion, we further calibrated the MSOT phantom images at different tissue thickness by multiplying with specific correction coefficients. As shown in Fig. R15a, by comparing with the OA images without pork tissues (as control), the phantom images of UCNs nanoprobe at 680 nm and 800 nm presented similar OA signals after correction, respectively. The ratiometric signals in Fig. R15c also showed comparable values after depth correction when the thickness was more than 8 mm. These results demonstrated the great feasibility and reliability of our UCNs nanoprobe for ratiometric mapping the dynamics of ROS and RNS stresses with a superior penetration depth in living settings.

Fig. R15. (a) MSOT phantom images of UCN (1 mg mL^{-1}) at 680 nm and 800 nm in the presence of the pork tissues at different depth with or without calibration. Scale bar: 1 mm. (b, c) the OA signals (b) and ratiometric values (c) of UNC's at 680 nm and 800 nm in different thickness.

Comment 5:

[(5) APAP stimulates ROS (e.g., Shuhendler et al) but the authors' OA data here suggest otherwise (and it is confusing to interpret 4d and 4f together). The presumption is that the 680 signal is killed by RNS? So is this really dual detection?]

Response:

We thank the reviewer's time and effort. First, sorry for the confusion caused here. As reported by Shuhendler *et al*, APAP stimulation induced the generation of ROS (e.g., H_2O_2), at initial 18 min. While, the prolonged administration of same dose of APAP at 53 min induced the significantly enhanced production of RNS (e.g., ONOO^-) in mice liver. In our studies, we demonstrated that APAP treatment displayed a slightly higher ROS (e.g., $\text{O}_2^{\cdot-}$) generation at initial 20 min and continuous increment of RNS (e.g., ONOO^-) production 60 min post-injection of APAP. Our results are actually very similar to the data reported by Shuhendler *et al*.

Additionally, sorry for the confusion in Fig. 4d and 4f. in our study, the RNS stimulation could specifically degrade Cy7 structure and decrease the very weak leading absorbance of Cy7 at 680 nm, which led to a slightly decreased OA signal at 680 nm observed by the reviewer. All our spectral analysis confirmed that RNS treatment could not affect HCy5 molecule, which can only be activated by ROS stimulation and resulted in an enhanced OA signal at 680 nm, opposite from the signal change caused by RNS-responsive Cy7. Herein, we sincerely appreciate the reviewer again for the valuable suggestion to effectively differentiate ROS/RNS contribution in complex mixture by the powerful mathematical deconvolution methods. By following the reviewer's comments, we performed the spectral deconvolution and orthogonally separate the specific contributions of ROS and RNS in their combinations to the OA signals changes at 680 nm and 800 nm in buffers, in cells and in living animals respectively, as shown in Fig. R9-11 in comment 1. We have also included these important information in our revised manuscript and supporting information.

Comment 6:

[(6) Protein induction of CYP450 by 30% in 7 minutes is surprisingly fast — please elaborate on how to interpret this finding.]

Response:

We apologize for the confusion caused here. Actually, in this study, in order to induce the hepatotoxicity, the mice were first intraperitoneally injected with INH (e.g. 200 mg Kg^{-1}) for 15 min, then followed by *i.v.* injection of UCN's probe for MSOT imaging and metabolism analysis at different time intervals. The assessment of metabolism CYP450 observed by the reviewer (in Fig. 6c or Fig. R16a) was analyzed at ~ 25 min after INH pre-treatment rather than the time at 7 min. The excessively hepatotoxic INH stimulation for 25 or 30 min may likely induce the obvious CYP450 protein expression as reported previously (e.g., *Drug Metab. Dispos.*, 2014, 42, 492; *Hepatology*, 2014, 59, 1084; *Drug Metab. Rev.*, 2015, 47, 222, etc).

Fig. R16. Assessment of metabolism-related enzymes including CYP450 (a) and CYP2E1 (b) at different time in mouse liver upon overdose INH (200 mg Kg⁻¹) treatment.

Moreover, such INH-triggered CYP450 protein induction was also re-confirmed by assessment of another CYP isoform (CYP2E1), a well-known central pathway for the ROS production and hepatotoxic injury (e.g., *Drug Metab. Rev.*, 2015, 47, 222; *Hepatology*, 2003, 37, 924; *Hepatology*, 2014, 59, 1084, etc). As shown in Fig. R16b, the results further proved the enhanced protein level of CYP2E1 (~ 35 %) after *i.p.* injection of INH within the time duration ~ 25-30 min, demonstrating the great possibility of CYP450 protein expression increment in mice liver at the early stage upon INH treatment.

Comment 7:

[(7) The authors use TUNEL as a marker of “DNA fragmentation associated with the ROS-induced cell damage”. More selective markers of ROS-mediated (more than RNS-mediated) damage should be added, since many processes can lead to TUNEL-positivity. TUNEL is good to keep however as a marker of downstream cell response.]

Response:

Thanks for the reviewer’s valuable comments. As suggested by the reviewer, we added another more specific biomarker, 4-hydroxynonenal (4-HNE), well-recognized as a hallmark of ROS overproduction, to determine the ROS-mediated liver injury. As shown in Fig. R17, the histological analysis presented negligible change in liver tissue within 60 min treatment of APAP or INH, and slightly more production of 4-HNE-positive foci after 180 min drugs stimulation, suggesting the occurrence of weak oxidative stress after overdose APAP or INH treatment. However, remarkable 4-HNE-positive lesion were easily observed at 60 min and 180 min upon overdose THA treatment, clearly indicating the ROS-mediated tissue damage during the process of THA metabolism, which were also comparable with the results reported previously (e.g., *Toxicol. In Vitro*, 2014, 28, 667; *Curr. Top Med. Chem.*, 2017, 17, 1006; *Chem. Biol. Drug Des.*, 2016, 87, 101, etc). The details have been included in our revised manuscript.

Fig. R17. Immunohistochemical analysis of 4-hydroxynonenal (4-HNE) staining in liver sections at 60 and 180 min after UCN, APAP, INH and THA treatment ($n = 5$). Arrowheads mark 4-HNE-positive lesions (black). Scale bars, 100 μm .

Comment 8:

[(8) Inline with the above comment, it would be good to have an ROS-signature treatment to compare with in Fig 6c-h (such as LPS, or early 60 min THA?). S18 has no time point labeled.]

Response:

We appreciate the reviewer's comments. By following the reviewer's suggestions, we examined the LPS-mediated inflammation procession by monitoring of different hepatotoxic factors in mouse liver, including metabolism-mediated enzymes (e.g., CYP450 and UGTs), pro-inflammatory cytokines (e.g., IL-6), histological and immunohistochemical analysis (e.g., H&E, 4-HNE, 3-nitrotyrosine). As shown in Fig. R18a-b, compared to THA, a similar CYP450 level along with a minimum suppression of UGTs activities were determined upon overdose LPS (20 mg Kg⁻¹) treatment for 60 min, suggesting the possibility of phase I but not phase II metabolism pathways occurred in LPS-induced liver injury. Moreover, a much significant enhancement of IL-6 was observed from 60 min to 120 min (Fig. R18c), indicating the continuous inflammation process under LPS stimulation. These results suggested that LPS could induce hepatotoxicity mainly based on the phase I metabolism in liver cells along with hepatotoxic cytokines generation, which are consistent with LPS-induced hepatotoxicity reported previously (e.g., *Hepatology*, 2017, 66, 953; *J. Endotoxin. Res.*, 2002, 8, 319; *Am. J. Physiol. Gastrointest. Liver Physiol.*, 2002, 283, G256, etc).

Fig. R18. The pathophysiological implications in LPS-induced inflammations procession *in vivo*. (a, b) the metabolism-related enzymes expression including CYP450 (a) and UGTs (b) at different time in mouse liver upon LPS treatment. (c) Time-resolved variations of inflammation-associated cytokines (IL-6) upon LPS administration. Statistical significance assessed by a Student's t-test (heteroscedastic, two-sided). * $P < 0.05$, ** $P < 0.01$, *** $P < 0.001$. (d) H&E staining in liver tissues at 180 min after UCN and LPS treatment. Arrowheads mark centrilobular vein fibrosis (blue), swollen hepatocytes (green) and inflammatory infiltration (red) respectively. CV: central vein. Scale bars, 50 μm . (e, f) Immunohistochemical analysis of 4-HNE (e) and 3-nitrotyrosine (f) staining in liver section at 60 and 180 min after UCN, LPS, APAP, INH and THA treatment ($n = 5$). Arrowheads mark 4-HNE-positive lesions (black) and nitrotyrosine-positive foci (white) respectively. Scale bars, 100 μm .

As suggested, we also performed the liver tissue histological and immunohistochemical analysis to evaluate the ROS-signature in inflammatory response upon overdose LPS treatment. As shown in Fig. R18d, the H&E staining indicated a minimum hepatotoxicity within 180 min after UCNs nanoprobe treatment. However, after LPS injection (20 mg Kg⁻¹) in living mice, the typical lobular hepatocyte structures were dramatically destroyed. The disparate histologic changes, such as centrilobular vein fibrosis, swollen hepatocytes, sinusoidal congestion and inflammatory infiltration, were readily observed up to 180 min. These studies clearly suggested the definite hepatotoxicity occurred later than the systematic drug metabolism and initial inflammation reaction, which is the process well-accepted in drug induced liver injury (DILI) (e.g., *J. Hepatol.*, 2015, 63, 503; *Cell Death Dis.*, 2015, 6, e1887; *Curr. Opin. Allergy Clin. Immunol.*, 2014, 14, 286, etc).

In order to validate the ROS-signature during the inflammation procession upon LPS treatment, 4-HNE, a specific and well-known hallmark of ROS overproduction, was also used to examine the ROS-induced cell damage. Moreover, as a product of tyrosine nitration in protein, 3-nitrotyrosine has been considered as alternative specific biomarker of nitrosative stress in living system. As shown in Fig. R18e, compared with other drugs, after LPS treatment, the histological changes in liver tissue presented obvious 4-HNE-positive foci at 60 min and 180 min, suggesting the occurrence of ROS-signature during the process of LPS-induced liver injury. Moreover, no obvious 3-nitrotyrosine staining foci was determined in liver tissue at 60 min and 180 min (Fig. R18f), demonstrating the major roles of LPS-triggered hepatotoxicity should be attributed to ROS instead of RNS after LPS treatment, which were also consistent with the LPS-induced liver injury reported previously (e.g., *Hepatology*, 2017, 66, 953; *Am. J. Physiol. Gastrointest. Liver Physiol.*, 2005, 289, G308; *Free Radic. Biol. Med.*, 2014, 73, 51, etc). Regrettably, considering the enormous data in our studies but limited space of manuscript, we only presented these results in the response letter.

Finally, we apologize to miss the time points of data collection in Fig. S18, which have been included in our revised supplementary information.

Comment 9:

[(9) Particle stability should be tested in media and ideally tissue homogenate.]

Response:

According to the reviewer's suggestions, we evaluated the stabilities of UCNs nanoprobe by monitoring of HCy5 and Cy7 absorption spectra (Fig. R19a) after incubation with cell culture medium (DMEM) and liver tissue homogenate. As shown in Fig. R19b-c, there was no significant absorbance change for Cy7 at 780 nm and HCy5 at 385 nm in DMEM (Fig. R19b), and in tissue homogenate (Fig. R19c), which are comparable with the relevant stability tests reported before (e.g., *ACS Nano*, 2016, 10, 10049; *Angew. Chem. Int. Ed.*, 2013, 52, 10325; *Adv. Funct. Mater.*, 2015, 25, 2386, etc).

Together with the studies discussed earlier, the results indicated that the Cy7/HCy5-loaded UCNs can act as a stable nanoplatform in physiological environment for further studies *in vivo*.

Fig. R19. (a) Absorbance spectra of Cy7, HCy5 and UCNs probe; (b, c) Stability tests of loaded Cy7 and HCy5 molecules on UCNs surface at 780 nm and 385 nm upon incubation with DMEM (b) and liver tissue homogenate (c) for different time.

Comment 10:

[(10) Longer term toxicity to UCN should be tested, especially since PEI is used which has known problems, especially branched PEI as used here. Nanomedicine (Lond). 2014 Feb; 9(2): 295–312.]

Response:

We thank the reviewer for such a valuable comment. We agree with the reviewer that the branched PEI could be a potential concern after longer term administration. So in our studies, we actually modified the PEI-conjugated UCNs with another biocompatible polymer, polyethylene glycol acid (PEG₅₀₀₀-COOH), on the surface of PEI-UCNs through the amide condensation reaction to eliminate the potential hepatotoxicity in living mice. As suggested by reviewer, we investigated the longer-term toxicity of PEI-UCNs and PEG/PEI-UCNs through the histological and hepatotoxic biomarker analysis. As shown in Fig. R20a-c, PEI-UCNs animal treatment for 14 days could lead to some certain pathological changes, indicating the possibility of longer term toxicity after prolonged PEI-nanoparticle administration. While, there was no obvious histological changes detected and our H&E staining indicated negligible hepatotoxicity within 14 days treatment of PEG/PEI-UCNs (5 mg/mL in 100 μ L saline), clearly suggesting the good biosafety of our proposed nanoplatform. These results suggested the possibility of PEG modification to effectively eliminate the long-term hepatotoxicity of PEI-conjugated nanoprobe, which was consistent with those reported previously (*e.g.*, *Scientific Report*, 2018, 8, 2082; *Small*, 2015, 11, 5066; *J. Control. Release.*, 2010, 144, 75, etc).

Fig. R20. (a) H&E staining of various organs upon PEG/PEI-UCN and PEI-UCN treatment including heart, lung, spleen and kidney. These organs were harvested at 0, 7 and 14 days after nanoparticle *i.v.* injection in living mice. Arrowheads mark centrilobular vein fibrosis (blue), swollen hepatocytes (green) and inflammatory infiltration (red) respectively. CV: central vein. Scale bars: 200 μ m. (b, c)

Quantification of ALT (**b**) and AST (**c**) biomarkers variations in mice liver at different days upon PEG/PEI-UCN and PEI-UCN administration ($n = 5$).

Minor points:

Comment 11:

[What is shown in 2d that is not apparent in 2e?]

Response:

We thank the reviewer to take time and read our manuscript in such a careful manner. By following the reviewer's suggestion, we have revised these two figures by utilizing the same ratiometric value calculation to differentiate the ROS and RNS.

Comment 12:

[Please better explain what the circles are besides figures 2e, 3b (presumably wells from a 96-well plate but where are the cells? what is the scale? etc); can't absolute values be used on all heat map images?]

Response:

We apologize for the confusion caused here. The circles with pseudo-color in Fig. 2e and 3b were the MSOT phantom images of UCNs nanoprobe in buffer solutions or in living cells. Typically, in our MSOT experiments, the OA signals in buffer or in living cells were not performed in a 96-well plate. The OA studies were actually carried out by encapsulating the solutions into a PA phantom containing two-channel polyurethane cylindrical (one for control and the other for sample). The PA scanning was recorded using a 128-element concave transducer array with the optimal excitation wavelength at 680-980 nm. Here, as suggested by the reviewer, we have included the scale bar and absolute values of these pseudo-color images in the revised manuscript and supporting information.

Comment 13:

[Groups 1-6 could be labeled embedded in the figure, it is a little inconvenient to find what the numbers correspond to in the caption.]

Response:

By taking the reviewer's suggestion, we have included the detailed conditions of group 1-6 in the Fig. 3 of revised manuscript.

Comment 14:

[Fig S8: the spectra should extend to 600nm as in Fig. S7.]

Response:

We thank the reviewer's suggestion here. By following the reviewer's comment, we extended the absorption spectra boundary in Fig. S8 to ~ 600 nm as in Fig. S7. Considering the very different excitation of RNS-responsive Cy7 molecule at 770 nm and ROS-regenerated Cy5 (from HCy5) structure at 630 nm, we presented the specific emission spectra of Cy5 and Cy7 fluorophores at different wavelength range in the revised supplementary information.

Comment 15:

[INA and THA are capable of inducing oxidative or nitrosative stress (should be INH).]

Response:

Sorry for the typo here. We replaced “INA” with “INH” in the revised manuscript.

Comment 16:

[Where is control group in 6b?]

Response:

As suggested by reviewer, we have investigated the ratiometric signal variations of UCNs nanoprobe as a control group in the revised manuscript (Fig. 6b).

Comment 17:

[Sample size, error bar definition, p-value, and statistical test used are missing in many panels.]

Response:

We thank the reviewer for such valuable suggestion. By following the reviewer’s comments, we have included the specific description of sample size, error bar definition, *p*-value, and statistical tests in the appropriate positions in the figures and captions of our revised manuscript and supporting information.

REVIEWERS' COMMENTS:

Reviewer #1 (Remarks to the Author):

The authors have made point-by-point response to the comments of all reviewers. The data added in revised manuscript have improved the publication and further supported the results of the authors. The revised manuscript has been polished and can help the readers understand the proposed method more easily. So the manuscript can be accepted for publication in Nature Communication.

Reviewer #3 (Remarks to the Author):

The authors did a great job in addressing my previous comments. The work has now been much improved and I have no more further concerns.

Reviewer #4 (Remarks to the Author):

The authors did a nice job addressing the first round of comments. It would be preferable to include more of the data in the supplement, much of which at present are only included in the point-by-point response. Some of the data are hinted at in other supplemental figures (R13 and R14, for instance). However, others (R15) are not?

R15 (shows 8mm depth is important to interpret Fig. 4-5). I did not see where in the manuscript this information was included, or why it wouldn't be crucial to consider when interpreting imaging in Fig. 4-5 (where diameter of tissue is >2 cm)?

Point-by-Point Response letter

The followings are our point-by-point response to the comments of reviewers and the changes made to the manuscript.

Reviewer #1 (Remarks to the Author):

The authors have made point-by-point response to the comments of all reviewers. The data added in revised manuscript have improved the publication and further supported the results of the authors. The revised manuscript has been polished and can help the readers understand the proposed method more easily. So the manuscript can be accepted for publication in Nature Communication.

Response:

We sincerely appreciate the reviewer's positive comments on our revised manuscript.

Reviewer #3 (Remarks to the Author):

The authors did a great job in addressing my previous comments. The work has now been much improved and I have no more further concerns.

Response:

We greatly thank the reviewer's time and effort in reviewing our revised manuscript.

Reviewer #4 (Remarks to the Author):

The authors did a nice job addressing the first round of comments. It would be preferable to include more of the data in the supplement, much of which at present are only included in the point-by-point response. Some of the data are hinted at in other supplemental figures (R13 and R14, for instance). However, others (R15) are not?

R15 (shows 8mm depth is important to interpret Fig. 4-5). I did not see where in the manuscript this information was included, or why it wouldn't be crucial to consider when interpreting imaging in Fig. 4-5 (where diameter of tissue is >2 cm)?

Response:

We sincerely appreciate the reviewer for such valuable comments on our manuscript. By following the reviewer's suggestions, we have included more data in the revised supplementary information to further strengthen our studies (e.g., Supplementary Fig. 11 and Fig. 26). Especially, we greatly agree with the reviewer that 8-mm penetration depth is important to interpret Fig. 4-5, so we have included the Fig. R15 in the revised Supplementary Fig. 11, and also presented the relevant information in the revised manuscript.